# Hippocampal ensemble dynamics and memory performance are modulated by respiration during encoding

Nozomu H. Nakamura [1] ✉, Hidemasa Furue[2], Kenta Kobayashi [3] & Yoshitaka Oku [1]

During offline brain states, such as sleep and memory consolidation, respiration coordinates hippocampal activity. However, the role of breathing during online memory traces remains unclear. Here, we show that respiration can be recruited during online memory encoding. Optogenetic manipulation was used to control activation of the primary inspiratory rhythm generator Pre-Bötzinger complex (PreBötC) in transgenic mice. When intermittent PreBötC-induced apnea covered the object exploration time during encoding, novel object detection was impaired. Moreover, the mice did not exhibit freezing behavior during presentation of fear-conditioned stimuli (CS⁺) when PreBötC-induced apnea occurred at the exact time of encoding. This apnea did not evoke changes in CA3 cell ensembles between presentations of CS⁺ and conditioned inhibition (CS⁻), whereas in normal breathing, CS⁺ presentations produced dynamic changes. Our findings demonstrate that components of central respiratory activity (e.g., frequency) during online encoding strongly contribute to shaping hippocampal ensemble dynamics and memory performance.

Breathing is an action that we take for granted, and until recently, few would have considered that breathing may directly affect our brain and change our memory traces. Breathing oscillations, which cause biphasic inspiratory–expiratory rhythms, comprise networking between excitatory and inhibitory neurons in the PreBötzinger complex (PreBötC) and Bötzinger complex in the ventrolateral medulla[1–5]. While such breathing patterns are altered by activating cognitive processes via the prefrontal cortex in a top-down manner[6–8], the respiratory system through the nasal cavity plays an important role in coupling neural oscillatory activity (i.e., delta and theta oscillations) in the olfactory bulb, barrel cortex, prefrontal cortex, and hippocampus in a bottom-up manner[9,10]. In particular, respiration-coupled brain oscillations may be critical for the occurrence of gamma oscillations, which are involved in cognitive function[10].

Recent human studies have demonstrated that the timing of respiratory cycles changes cognitive performance[11–15]. In a cognitive task, accuracy was enhanced when presentations of test cues started with an expiratory-to-inspiratory phase (EI) transition (equivalent to the onset time of inspiration) during the respiratory cycle[13]. However, accuracy was reduced when the EI transition occurred in the middle of the retrieval process[12,15]. The EI transition is generated by activation of the primary inspiratory rhythm generator PreBötC[2,3] and can be determined in an "abrupt or divergent manner", suggesting that EI transition-dependent signals may strongly modulate cognitive function.

Memories are stored through mechanisms of synaptic plasticity that change and endure the activity of cell ensembles in the brain, known as memory traces or memory engrams[16,17]. Karalis and Sirota[18] recently demonstrated that respiration coordinates the dynamics of

[1]Division of Physiome, Department of Physiology, Hyogo Medical University, 1-1, Mukogawa cho, Nishinomiya, Hyogo 663-8501, Japan. [2]Division of Neurophysiology, Department of Physiology, Hyogo Medical University, 1-1, Mukogawa cho, Nishinomiya, Hyogo 663-8501, Japan. [3]Section of Viral Vector Development, National Institute for Physiological Sciences, 38 Nishigonaka Myodaiji, Okazaki, Aichi 444-8585, Japan. ✉e-mail: no-nakamura@hyo-med.ac.jp

hippocampal high-frequency ripple oscillations[19] and prefrontal active states during "offline" brain states, such as sleep and memory consolidation. Hippocampal sharp-wave ripples are identified to correspond to the reoccurrence of a sequence of cell activation during consolidation, named hippocampal replay[20]. Thus, the respiratory modulation of population activity could explain the mechanism underlying the occurrence of hippocampal replay during offline consolidation[18]. Meanwhile, these replay sequences in the same or reverse order reflect hippocampal cell ensembles during memory encoding[20,21]. Even though certain respiration timing can be recruited in activating "online" memory processes in human studies[12,13,15], whether hippocampal cell ensembles involve respiratory activity during encoding remains unclear.

Here, we address the hypothesis that PreBötC-derived activity during online encoding modulates the dynamics of hippocampal cell ensembles. To test this hypothesis, we employed optogenetic manipulation to control the in vivo activation of the PreBötC[22], which periodically generates EI transitions. Then, we determined whether memory performance was changed by temporary apnea (or respiratory arrest) and degraded rhythmic activity derived from the PreBötC at the exact time of encoding. Using cellular compartment analysis of temporal activity by fluorescent in situ hybridization (catFISH) on the immediate-early genes (IEGs) *Arc*, *Fos*, and *Homer1a* (*H1a*)[23–26], we characterized the dynamics of cell ensembles in the hippocampus and prefrontal cortex[27] between presentations of conditioned stimuli (CS⁺) and the conditioned inhibition of fear (CS⁻). Our results demonstrate that the neural signature of hippocampal ensemble dynamics, which may underlie memory encoding and momentary links between information sequences, is stimulated by EI transition-dependent signals derived from the PreBötC.

## Results

### Regulation of breathing and optogenetic manipulation

According to Sherman and colleagues[22], optogenetic photostimulation activates inhibitory neurons in the PreBötC, which are specifically expressed with channelrhodopsin-2 (ChR2). Then, these inhibitory neurons immediately suppress excitatory neurons in the PreBötC, resulting in reduced respiratory frequency and short-term apnea. Adeno-associated viruses (AAVs), which encode ChR2-EYFP driven by the constitutive promoter Ef1a in a double loxP-flanked inverted open reading frame configuration (DIO-ChR2), were bilaterally injected into the PreBötC in transgenic mice expressing Cre recombinase under the promotor site of the vesicular GABA transporter (*Vgat*-Cre⁺ or Cre⁺ mice, Fig. 1a–c, see "Methods"). During isoflurane-induced anesthesia, the results of nonparametric Friedman test and post hoc pairwise comparisons showed that photostimulation (flat, 2s) completely inhibited the respiratory frequency in Cre⁺ mice (Fig. 1d, e). Side effects of the optogenetic manipulation on physiological responses, including very small changes in the R wave-to-R wave (RR) intervals between heartbeats (1.2–1.3%) measured by electrocardiogram in Cre⁺ mice under anesthesia, were observed (Fig. 1f). In the awake state, the one-way repeated-measures ANOVA showed no change in the frequency of whole-body plethysmographic signals in *Vgat*-Cre⁻ (wild-type or Cre⁻) mice; however, in Cre⁺ mice the frequency was markedly decreased by 90.7–91.1% (0.63 Hz) during the photostimulation period (flat, 2 s) compared to the frequency in the pre- and poststimulation periods (post hoc pairwise *t* test with two-sided and Bonferroni correction, Fig. 1g, h and Supplementary Figs. 1 and 2). These results indicated that apnea was driven immediately by photostimulation through optical fibers to the PreBötC in Cre⁺ mice but not in Cre⁻ mice.

### Object-recognition memory using optogenetic manipulation

We next investigated whether the exclusion of respiration during encoding disrupted memory performance. Rodents have a natural preference to explore a novel object more than a familiar object[28].

Based on this preference, we employed an object-recognition memory task[29] involving two 6-min epochs: sample and recognition epochs (Fig. 2a). During the sample epoch, animals spontaneously explored an open field with two identical objects, and the PreBötC was manually photostimulated by an investigator when the animals were located within 2 cm of each object (i.e., object exploration). Especially, photostimulation was intermittently generated with a 2-s duration even when animals maintained their position. After a 20-min delay, the animals again spontaneously explored the open field, and one of the objects was replaced by a novel object that the animals had never encountered before.

No differences were observed in total object exploration time (within a 2-cm distance) during the sample and recognition epochs between Cre⁻ and Cre⁺ mice (Supplementary Fig. 3a). The two-way mixed-design ANOVA and two-tailed paired *t* test showed that the discrimination ratio was higher during the recognition epoch than during the sample epoch in Cre⁻ mice as expected, whereas no difference in the ratio was observed in Cre⁺ mice (Fig. 2b). Notably, Cre⁻ mice had a higher discrimination ratio against zero (two-tailed one-sample *t* test) and a higher discrimination ratio than Cre⁺ mice during the recognition epoch (two-tailed Welch's *t* test). No correlations were found between the exposure time and coverage ratio of photostimulation during the sample epoch and the discrimination ratio during the recognition epoch in Cre⁺ mice (Fig. 2c, d and Supplementary Fig. 3b). As a result, 60.9% of the object exploration time was covered with intermittent photostimulation, which caused PreBötC-induced apnea in Cre⁺ mice (Fig. 2d). Thus, these results revealed that object-recognition memory was disrupted by intermittent PreBötC-induced apnea, which covered more than half of the object exploration time during encoding.

### Fear conditioning in an acrylic box using optogenetic manipulation

Previous studies have shown that breathing modulates olfactory-dependent discrimination and performance[30–32]. As rodents typically use olfactory strategies for object exploration, the performance deficits induced by apnea can be olfactory-dependent. To exclude such effects, we designed a fear-conditioning paradigm, in which each animal spent the entire time in a transparent acrylic box and then used a certain strategy without object or olfactory components (Fig. 3a). Each animal was placed into the acrylic box, which was away from either black or white walls. We utilized a pair of fear-conditioned stimuli (CS⁺) and conditioned inhibition (CS⁻) in the paradigm[33]. Several conditioning paradigms were preliminarily tested in combination with a box, different contexts, lights, and tones (i.e., Groups A-G, see "Methods", Supplementary Figs. 4–8). We found that the animals in the acrylic box were effectively conditioned to different combinations of contexts and tones, which were associated with footshocks, and used the paradigm of Group G for further experiments (i.e., CS⁻: black context and low tones; CS⁺: white context and high tones; see Supplementary Fig. 8).

Then, we examined whether short-term PreBötC-induced apnea occurring during encoding decreased subsequent memory performance. The paradigm consisted of 2 days for the sample sessions and 1 day for the recognition session (Fig. 3b), with two 5-min epochs (Epochs 1 and 2) per day (Fig. 3c). During the sample sessions, the Cre⁻ or Cre⁺ mouse was placed in the acrylic box surrounded by either black or white walls (Fig. 3d). White walls and high tones with photostimulation served as the CS⁺, and black walls and low tones without photostimulation served as the CS⁻ (see "Methods"). The animal was photostimulated twice for 4 s (flat) after high tones were presented (a total of 8 photostimulation periods during each sample session), the first of which completely covered the timing of a 2-s footshock (Fig. 3e). We confirmed that activity levels in the acrylic box were considerably higher during the footshock than during the prestimulation period

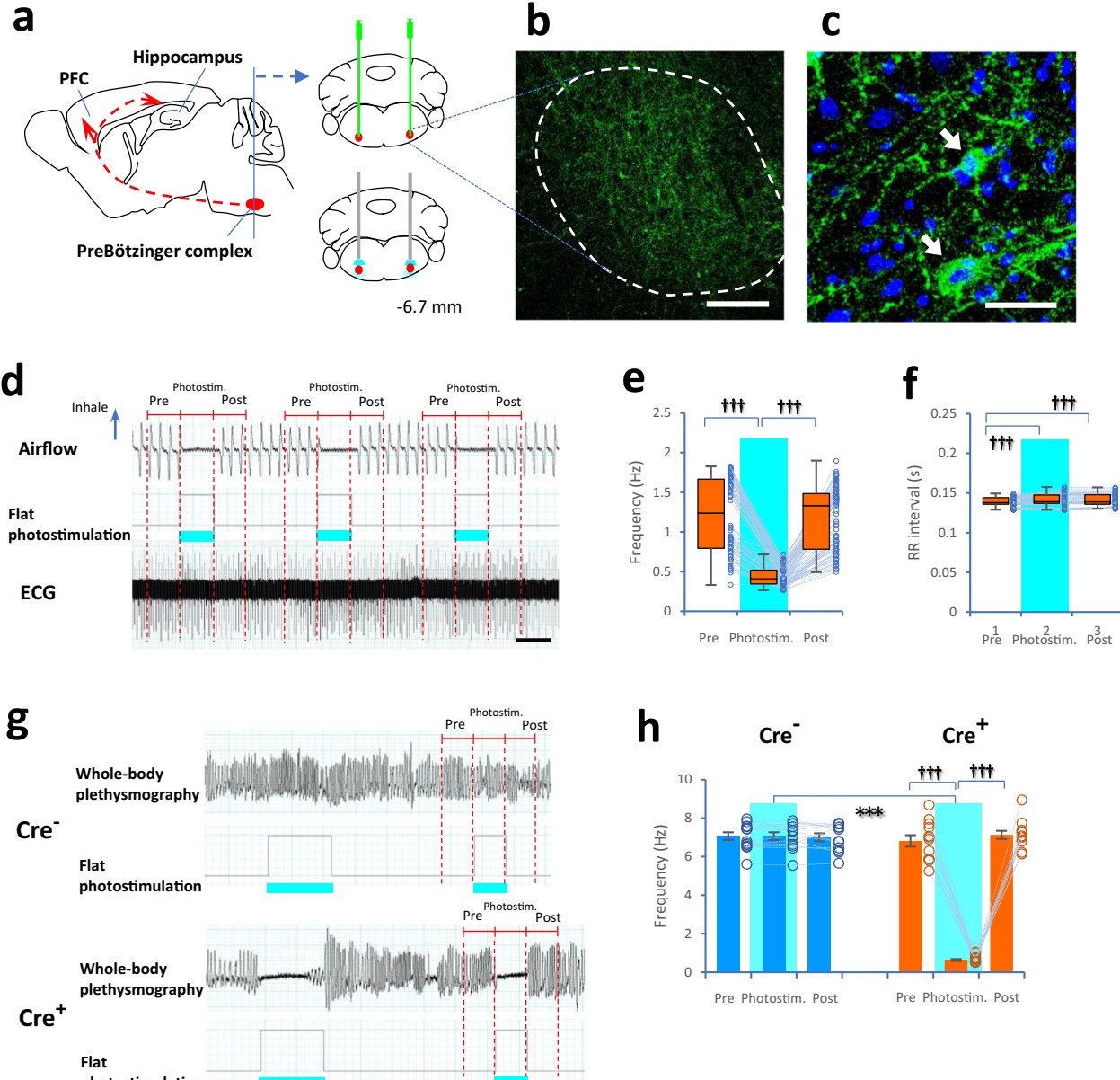

**Fig. 1 | Regulation of breathing and optogenetic manipulations. a** Drawings showing the PreBötzinger complex (PreBötC), prefrontal cortex, and hippocampus. *AAV-Ef1α-DIO-hChR2(H134R)-EYFP* was injected bilaterally into the PreBötC (6.70 mm caudal to bregma, red) in *Vgat*-Cre⁺ mice and fiber-optic cannulas were placed bilaterally in the dorsal region (adapted from ref. 60). **b, c** Images showing EYFP positive inhibitory Vgat neurons (green, arrows) and DAPI-stained nuclei (blue) in the PreBötC. Similar patterns of expression were confirmed independently at least five times. **d**–**f** Box plots showing the respiratory frequency (**d, e**) and RR intervals in electrocardiograms (**d, f**) during the prestimulation (2 s), photostimulation (flat, 465 nm as blue light, 2 s), and poststimulation (2 s) periods in anesthetized *Vgat*-Cre⁺ mice (20 trials per animal, $n = 4$ males; respiratory frequency: $\chi^2(2) = 120.33$, $P < 2.2 \times 10^{-16}$, Friedman test; photostimulation: $P = 2.5 \times 10^{-15}$ compared to prestimulation, $P = 2.4 \times 10^{-14}$ compared to poststimulation, post hoc pairwise Wilcoxon signed-rank test with two-sided and Bonferroni correction; RR interval: $\chi^2(2) = 68.03$, $P = 1.7 \times 10^{-15}$; prestimulation: $P = 2.9 \times 10^{-8}$ compared to photostimulation, $P = 1.1 \times 10^{-12}$ compared to poststimulation). **g, h** Bar plots

showing the frequency of whole-body plethysmographic signals during the prestimulation (2 s), photostimulation (blue light, 2 s), and poststimulation (2 s) periods in awake *Vgat*-Cre⁻ mice (Cre⁻, $n = 12$, 3 females and 9 males, blue; $F(2, 22) = 0.20$, $P = 0.8$, one-way repeated-measures ANOVA) and *Vgat*-Cre⁺ (Cre⁺, $n = 12$, 5 females and 7 males, orange; $P = 0.02$, Mauchly tests for sphericity; $F(2, 22) = 463.3$, $P < 2.2 \times 10^{-16}$ with Greenhouse–Geisser correction; photostimulation: $P = 2.3 \times 10^{-11}$ compared to poststimulation, $P = 1.9 \times 10^{-9}$ compared to prestimulation, post hoc pairwise $t$ test with two-sided and Bonferroni correction) mice. During the 2-s photostimulation period (sky blue), the frequency of whole-body plethysmographic signals was lower in Cre⁺ mice than in Cre⁻ mice ($t(12.54) = 31.47$, $P = 2.6 \times 10^{-13}$, two-tailed Welch's $t$ test, **h**). The scale bars are 200 µm in (**b**), 50 µm in (**c**), and 2 s in (**d, g**). †††$P < 0.005$ (post hoc pairwise comparisons), ***$P < 0.005$ (two-tailed Welch's $t$ test). Box plots indicate median, first, and third quantiles and minimum and maximum values. Bar plots indicate means ± SEM. Circles in the graph represent individuals.

(animal type: $F(1, 20) = 0.50$, $P = 0.5$; period: $F(1, 20) = 469.5$, $P = 2.3 \times 10^{-15}$; interaction: $F(1, 20) = 0.01$, $P = 0.9$, two-way mixed-design ANOVA; prestimulation vs. footshock: Cre⁻: $t(10) = 17.24$, $P = 9.1 \times 10^{-9}$; Cre⁺: $t(10) = 13.95$, $P = 7.0 \times 10^{-8}$, two-tailed paired $t$ test, Fig. 3f).

Importantly, no difference in the activity level of Cre⁻ and Cre⁺ mice was observed (footshock: $t(18.96) = 0.37$, $P = 0.7$, two-tailed Welch's $t$ test), even though Cre⁺ mice were not supposed to breathe during the footshock.

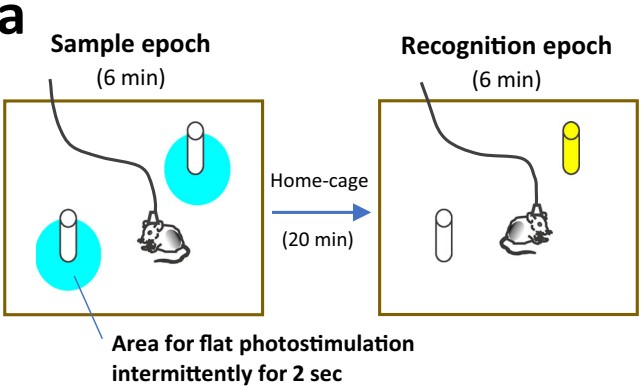

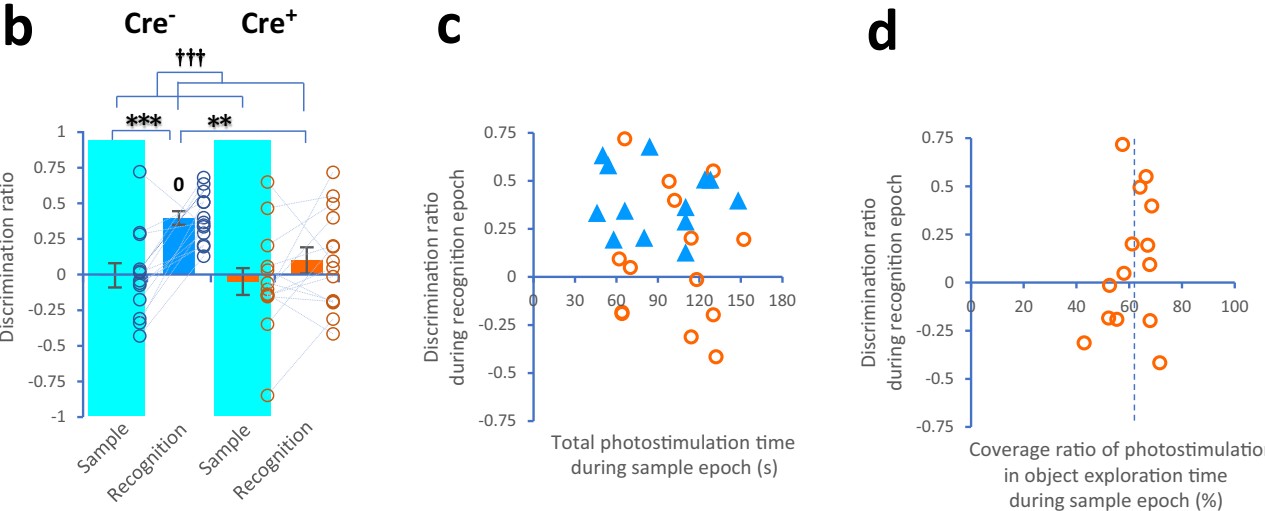

**Fig. 2 | Object-recognition memory using optogenetic manipulation.**
**a** Drawings showing two 6-min epochs, the sample epoch (left panel) and recognition epoch (right panel); positions for intermittent photostimulation for 2 s (sky blue); and a novel object (yellow) (adapted from ref. 33). **b** Bar plots showing the discrimination ratio (to the novel) during the sample and recognition epochs between *Vgat*-Cre⁻ (Cre⁻, blue) and *Vgat*-Cre⁺ (Cre⁺, orange) mice (animal type: $F_{(1, 25)} = 3.19$, $P = 0.09$; epoch: $F_{(1, 25)} = 15.54$, $P = 0.0006$; interaction: $F_{(1, 25)} = 3.44$, $P = 0.08$, two-way mixed-design ANOVA). The discrimination ratio was higher during the recognition epoch than during the sample epoch in Cre⁻ mice ($t_{(12)} = 4.45$, $P = 0.0008$, two-tailed paired *t* test) but not in Cre⁺ mice ($t_{(13)} = 1.45$, $P = 0.2$). During the recognition epoch, the discrimination ratio of Cre⁻ mice was higher than that of Cre⁺ mice ($t_{(19.47)} = 2.85$, $P = 0.010$, two-tailed Welch's *t* test; Cre⁻: $t_{(12)} = 8.22$, $P = 2.8 \times 10^{-6}$; Cre⁺: $t_{(13)} = 1.07$, $P = 0.3$, two-tailed one-sample *t* test against zero). **c** Plots showing no correlation between the total photostimulation time during the sample epoch and the discrimination ratio during the recognition epoch in Cre⁻ (blue) and Cre⁺ (orange) mice (Cre⁻: $r = -0.08$, $t_{(11)} = 0.26$, $P = 0.8$; Cre⁺: $r = -0.09$, $t_{(12)} = 0.30$, $P = 0.8$, Pearson's product-moment correlation). **d** Plots showing no correlation between the percentage coverage ratio of photostimulation during the object exploration time in the sample epoch ($60.9 \pm 2.2\%$) and the discrimination ratio during the recognition epoch in Cre⁺ mice ($r = 0.26$, $t_{(12)} = 0.92$, $P = 0.4$). Cre⁻ mice ($n = 13$, 7 females and 6 males, blue) and Cre⁺ mice ($n = 14$, 10 females and 4 males, orange) in (**b–d**). ⁺⁺⁺$P < 0.005$ (main effects and interactions using ANOVA). **$P \leq 0.01$ and ***$P < 0.005$ (two-tailed paired *t* test and two-tailed Welch's *t* test), ⁰$P < 0.005$ (two-tailed one-sample *t* test against zero). Bar plots indicate means ± SEM. Circles and triangles in the graph represent individuals.

Twenty-four hours later, we tested the ability of the animals to distinguish between the CS⁺ and CS⁻ combined with different contexts and tones (Fig. 3g). We estimated the temporal parameter of activity in the acrylic box (i.e., active time) during each onset and offset time block, instead of a conventional freezing index (Fig. 3h, i, see "Methods"). Two-way mixed-design ANOVA showed a significant main effect of epoch and a significant interaction between animal type and epoch during onset time blocks in Cre⁻ and Cre⁺ mice (offset: animal type: $F_{(1, 20)} = 1.20$, $P = 0.3$; epoch: $F_{(1, 20)} = 1.75$, $P = 0.2$; interaction: $F_{(1, 20)} = 3.90$, $P = 0.062$; onset: animal type: $F_{(1, 20)} = 2.51$, $P = 0.13$, epoch: $F_{(1, 20)} = 45.58$, $P = 1.4 \times 10^{-6}$; interaction: $F_{(1, 20)} = 20.51$, $P = 0.0002$, Fig. 3i). In Cre⁻ mice, presentation of the context/tone-dependent CS⁺ resulted in lower active time than presentation of the context/tone-dependent CS⁻ as expected ($t_{(10)} = 6.71$, $P = 0.00005$, two-tailed paired *t* test). Moreover, in Cre⁺

mice, no significant difference in active time was observed between the presentation of context/tone-dependent CS⁻ and CS⁺ ($t_{(10)} = 2.06$, $P = 0.07$). Furthermore, the active time of Cre⁺ mice was greater than that of Cre⁻ mice during the presentation of context/tone-dependent CS⁺ ($t_{(19.21)} = 2.83$, $P = 0.011$, two-tailed Welch's *t* test). Our results showed that memory performance was decreased by PreBötC-induced apnea, which occurred at the exact time of encoding (i.e., an association with footshocks).

**Immediate-early gene catFISH methods for hippocampal cell ensembles**
To determine whether memory encoding was modulated by central inspiratory activity at the molecular and cellular levels, we investigated the dynamics of cell ensembles in the hippocampus between the presentation of the CS⁻ and CS⁺ during the conditioning task. We

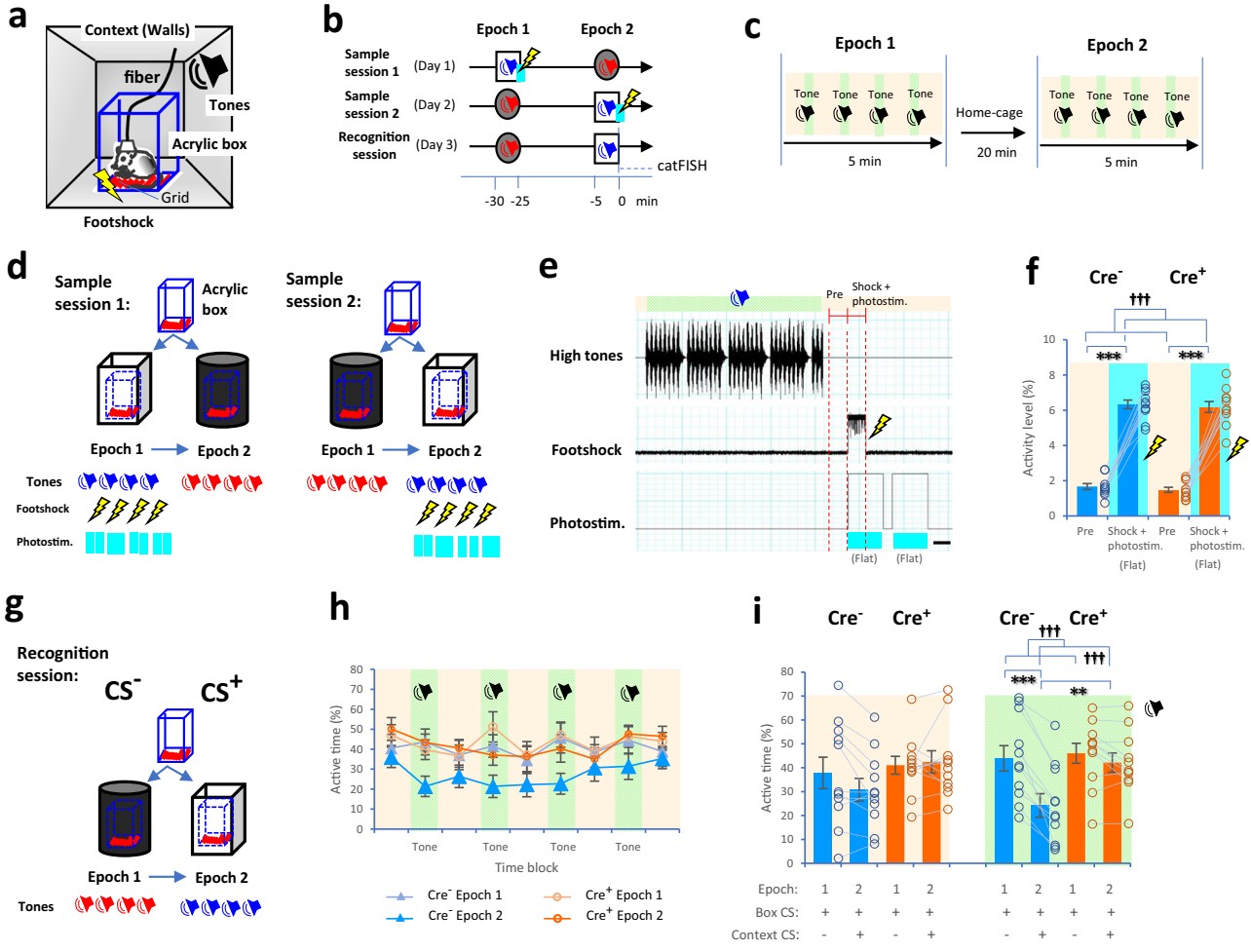

**Fig. 3 | Fear conditioning in an acrylic box using optogenetic manipulation.**
**a**, **b** Drawings (adapted from ref. 33) and diagrams showing a transparent acrylic box surrounded by walls (**a**) and the conditioning paradigm (**b**, see "Methods"). **c** Each epoch alternated between offset time blocks (light orange zone) and onset time blocks (tones, light green zone). **d**, **e** During the sample sessions, individual Cre⁻ and Cre⁺ mice were placed in the box and learned an association between white walls/high tones and footshocks under flat photostimulation exposure (sky blue) as the CS⁺ (**e**) and black walls/low tones without footshocks as the CS⁻. **f** Bar plots showing the activity levels of Cre⁻ (blue) and Cre⁺ (orange) mice in the

prestimulation period (2 s, **e**) and footshocks (2 s, **e**) during the sample session. **g** The recognition session for discriminating between the CS⁻ and CS⁺. **h**, **i** Plots showing the active time during the offset (light orange zone) and onset (light green zone) time blocks between the CS⁻ and CS⁺. The scale bar is 2 s in (**e**). Cre⁻ mice (n = 11, 3 females and 8 males, blue) and Cre⁺ mice (n = 11, 4 females and 7 males, orange) in (**f**, **i**). †††P < 0.005 (main effects and interactions using ANOVA), **P ≤ 0.01 and ***P < 0.005 (two-tailed Welch's t test and two-tailed paired t test). Bar and line plots indicate means ± SEM. Circles in the graph represent individuals.

employed catFISH methods with IEG *Arc*, *Fos*, and *Homer1a* (*H1a*) expression in the hippocampus[23,24,26] (Supplementary Fig. 9). The CA3 and CA1 of the hippocampus are functionally segregated along the transverse (proximodistal) axis[25,34]: distal CA3 (close to CA2) and proximal CA1 (close to CA2) are primarily involved in contextual memory, whereas proximal CA3 (close to dentate gyrus) and distal CA1 (close to subiculum) are preferentially involved in object (item) memory. Then, the dynamics of cell ensembles were examined in the distal and proximal parts of CA3 (Fig. 4a, b) and CA1 (Supplementary Fig. 10). Notably, the timing of Epoch 1 (i.e., CS⁻) in the conditioning paradigm was designed to fit "cytoplasmic *Arc*-positive" and "nuclear *H1a*-positive" signals, while the timing of Epoch 2 (i.e., CS⁺) was designed to fit "nuclear *Arc*-positive" and "nuclear *Fos*-positive" signals (see "Methods" and Supplementary Fig. 9).

First, we examined the proportion of "cytoplasmic *Arc*-positive" and "nuclear *Arc*-positive" cells in the distal CA3 (Dist. CA3, Fig. 4c–e). The percentage of nuclear *Arc*-positive cells was higher than the percentage of cytoplasmic *Arc*-positive cells in Cre⁻ mice (P = 0.03, two-sided Wilcoxon signed-rank test, Fig. 4f). However, no difference was

observed between nuclear and cytoplasmic *Arc*-positive cells in Cre⁺ mice (P = 0.09). Cre⁺ mice had a higher percentage of cytoplasmic *Arc*-positive cells than Cre⁻ mice (P = 0.03, two-sided Mann–Whitney U test). In contrast, no difference in the percentage of nuclear *Arc*-positive cells was observed between Cre⁻ and Cre⁺ mice (P = 0.4). Second, regarding the cells coexpressing cytoplasmic and nuclear *Arc*-positive signals (i.e., reactivation of the cells during Epoch 2 that were initially activated during Epoch 1) in the distal CA3, the percentage of cells coexpressing *Arc*-positive signals was higher than chance (i.e., cytoplasmic *Arc* x nuclear *Arc*) in Cre⁺ mice (P = 0.03, two-sided Wilcoxon signed-rank test), but not in Cre⁻ mice (P = 0.2, Fig. 4g). Moreover, the percentage of cells coexpressing *Arc*-positive signals was higher in Cre⁺ mice than in Cre⁻ mice (P = 0.03, two-sided Mann–Whitney U test). Third, in the proximal CA3 (Prox. CA3), the pattern proportion of *Arc*-positive cells was similar to but appeared lower than that in the distal CA3 (cytoplasm vs. nucleus in Cre⁻: P = 0.03, two-sided Wilcoxon signed-rank test; Cre⁺: P = 0.2; Cre⁻ vs. Cre⁺ in the cytoplasm: P = 0.03, two-sided Mann–Whitney U test; nucleus: P = 0.3, Fig. 4h; chance vs. real in Cre⁻: P = 0.11, two-sided Wilcoxon signed-rank test; Cre⁺:

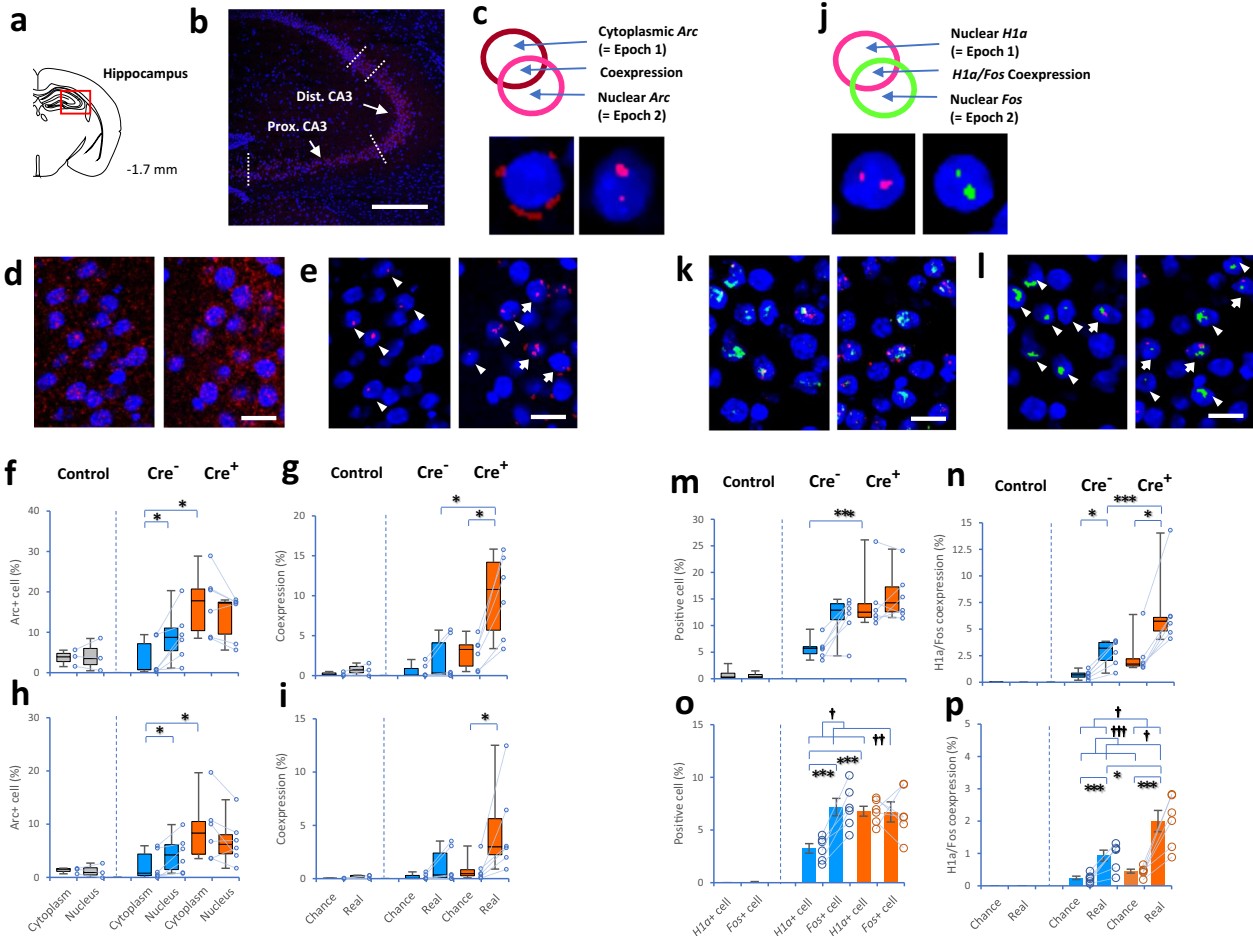

**Fig. 4 | IEG catFISH methods for assessing hippocampal CA3 cell ensembles.**
**a**, **b** Drawings (adapted from ref. 60) and images showing the dorsal hippocampus (**a**) and CA3 with DAPI-stained nuclei (blue, **b**). **c** Diagrams showing that cyto-plasmic *Arc* expression alone (dark red) reflects activation of Epoch 1; nuclear *Arc* expression alone (pink) reflects activation of Epoch 2; and cytoplasmic *Arc*/nuclear *Arc* coexpression reflects reactivation of cells. **d**, **e** Images showing cells expressing *Arc* in the distal part of CA3 in Cre⁻ (left panel) and Cre⁺ (right panel) mice before (**d**) and after (**e**) data processing. Cells coexpressing cytoplasmic and nuclear *Arc*-positive signals (arrows), and cells expressing nuclear *Arc* alone (arrowheads). **f–i** Box plots showing the proportion of cells expressing cytoplasmic *Arc*-positive and nuclear *Arc*-positive signals in the distal part of CA3 (Dist. CA3, **f**, **g**) and proximal part of CA3 (Prox. CA3, **h**, **i**) among home-cage control (gray), Cre⁻ (blue) and Cre⁺ (orange) mice (Supplementary Figs. 9 and 10). **j** Diagrams showing that nuclear *H1a* expression alone (pink) reflects activation of Epoch 1; nuclear *Fos* expression alone (green) reflects activation of Epoch 2; and *H1a/Fos* coexpression reflects reactivation of cells. **k**, **l** Images showing the cells expressing *H1a* (pink) and

*Fos* (green) positive signals in distal CA3 in Cre⁻ (left panel) and Cre⁺ (right panel) mice before (**k**) and after (**l**) data processing. Cells coexpressing *H1a* and *Fos*-positive signals (arrows), and cells expressing nuclear *Fos* alone (arrowheads). **m–p** Plots showing the proportion of *H1a* and/or *Fos*-positive cells in the distal (**m**, **n**) and proximal (**o**, **p**) parts of CA3 among home-cage control (gray), Cre⁻ (blue) and Cre⁺ (orange) mice. The scale bars are 300 μm in (**b**), 20 μm in (**d**, **e**, **k**, **l**). Home-cage control mice (*n* = 3, 1 female and 2 males in (**f**, **g**); *n* = 4, 1 female and 3 males in (**m–p**), gray), Cre⁻ mice (*n* = 6, 2 females and 4 males, blue) and Cre⁺ mice (*n* = 6, 2 females and 4 males, orange) in (**f–i**, **m–p**). †*P* < 0.05, ††*P* ≤ 0.01 and †††*P* < 0.005 (main effects and interactions using ANOVA). *P* < 0.05 and ***P* ≤ 0.005 (two-tailed Welch's *t* test, two-tailed paired *t* test, two-tailed Mann–Whitney *U* test, and two-tailed Wilcoxon signed-rank test). Similar patterns of IEG expression in the images were confirmed independently at least five times. Box plots indicate median, first, and third quantiles, and minimum and maximum values. Bar plots indicate means ± SEM. Circles in the graph represent individuals.

*P* = 0.03; Cre⁻ vs. Cre⁺ in coexpression: *P* = 0.09, two-sided Mann–Whitney *U* test, Fig. 4i).

Fourth, we found that the differences in the proportion of "nuclear *H1a*-positive" and "nuclear *Fos*-positive" cells in the distal CA3 (Fig. 4j–l) were similar to those in the proportion of cytoplasmic *Arc*-positive and nuclear *Arc*-positive cells (*H1a* vs. *Fos* in Cre⁻: *P* = 0.063, two-sided Wilcoxon signed-rank test; Cre⁺: *P* = 0.7; Cre⁻ vs. Cre⁺ in *H1a*: *P* = 0.002, two-sided Mann–Whitney *U* test; *Fos*: *P* = 0.3, Fig. 4m; chance vs. real in Cre⁻: *P* = 0.03, two-sided Wilcoxon signed-rank test; Cre⁺: *P* = 0.03; Cre⁻ vs. Cre⁺ in coexpression: *P* = 0.002, two-sided Mann–Whitney *U* test, Fig. 4n). In the proximal CA3, the pattern population of *Fos*- and *H1a*-positive cells was similar to but appeared lower than that in the distal CA3 (animal type: *F*(1, 10) = 4.08, *P* = 0.07; gene-type: *F*(1, 10) = 8.56, *P* = 0.02; interaction: *F*(1, 10) = 9.21, *P* = 0.013,

two-way mixed-design ANOVA; *Fos* vs. *H1a* in Cre⁻: *t*(5) = 5.99, *P* = 0.002; Cre⁺: *t*(5) = 0.06, *P* = 0.95; Cre⁻ vs. Cre⁺ in *H1a*: *t*(9.99) = 5.40, *P* = 0.0003; nucleus: *t*(9.80) = 0.37, *P* = 0.7, Fig. 4o; animal type: *F*(1, 10) = 8.68, *P* = 0.0146; coexpression: *F*(1, 10) = 48.66, *P* = 0.00004; interaction: *F*(1, 10) = 7.23, *P* = 0.02; chance vs. real in Cre⁻: *t*(5) = 4.76, *P* = 0.0051; Cre⁺: *t*(5) = 5.41, *P* = 0.003; Cre⁻ vs. Cre⁺ in coexpression: *t*(7.50) = 2.87, *P* = 0.02, Fig. 4p). These results revealed that the dynamics of CA3 cell ensembles for memory function were disrupted by PreBötC-induced apnea during the exact time of encoding.

**Correlation between neuronal activation and behavioral factors**
To further confirm the reliability of interactions between behavioral factors and neuronal activation during memory performance, we ana-lyzed correlations between changes in active time (tone CS⁺ – tone CS⁻)

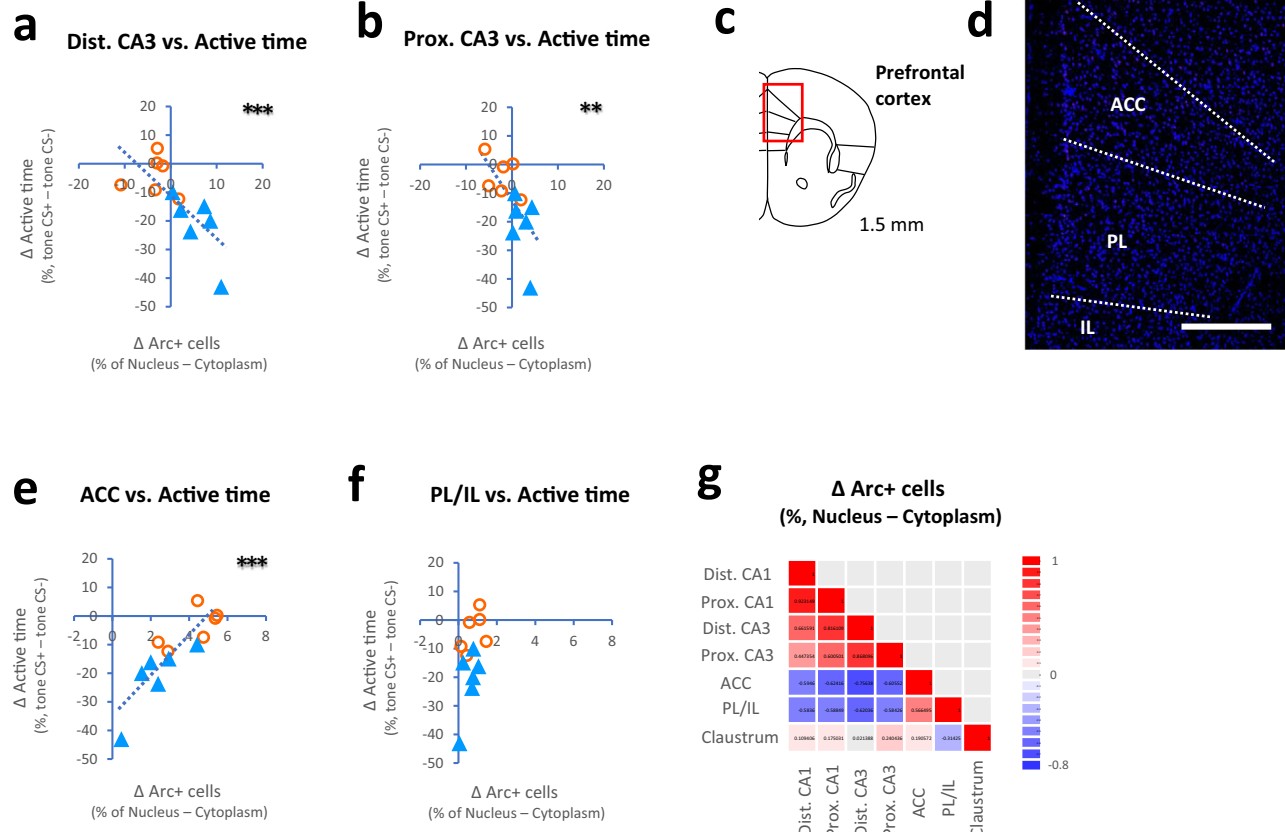

**Fig. 5 | Correlation between neuronal activation and behavioral factors.**
**a**, **b** Plots showing negative correlations between changes in active time (tone
CS$^+$–tone CS$^-$) and changes in the percentage of *Arc*-positive cells
(nucleus–cytoplasm) in the distal CA3 (**a**) and proximal CA3 (**b**) in Cre$^-$ (blue) and
Cre$^-$ (orange) mice. **c**, **d** Drawings (adapted from ref. 60) and images showing the
medial prefrontal cortex. **e**, **f** Plots showing correlations between changes in the
percent active time (tone CS$^+$–tone CS$^-$) and changes in the percentage of *Arc*-

positive cells (nucleus–cytoplasm) in the anterior cingulate cortex (ACC, **e**) and
prelimbic/infralimbic cortex (PL/IL, **f**) in Cre$^-$ (blue) and Cre$^-$ (orange) mice.
**g** Columns showing the correlation coefficients of changes in the percentage of *Arc*-
positive cells in the hippocampus and prefrontal cortex. The scale bar is 400 μm in
(**d**). **P ≤ 0.01, and ***P < 0.005 (Pearson's product-moment correlation). Circles
and triangles in the graph represent individuals.

and changes in the percentage of *Arc*-positive cells (nucleus–cytoplasm)
in the hippocampus and prefrontal cortex. Interestingly, negative cor-
relations between changes in active time and changes in the *Arc*-positive
cell population were observed in both distal and proximal parts of CA3
(distal CA3: $r = -0.76$, $t(10) = 3.64$, $P = 0.0045$, Pearson's product-
moment correlation, Fig. 5a; proximal CA3: $r = -0.69$, $t(10) = 2.98$,
$P = 0.014$, Fig. 5b). While a positive correlation was observed in the
anterior cingulate cortex (ACC: $r = 0.86$, $t(10) = 5.40$, $P = 0.0003$,
Fig. 5c–e), no correlation was found in the prelimbic/infralimbic cortex
(PL/IL: $r = 0.53$, $t(10) = 1.98$, $P = 0.08$, Fig. 5f), claustrum, or CA1 (Sup-
plementary Fig. 11 and Fig. 5g). We found that following optogenetic
manipulation, the pattern activities in the hippocampus and prefrontal
cortex were maintained differentially as representations of memory
performance and subsequent motivated behavior, respectively.

**Fear conditioning using optogenetic manipulation and 10-Hz
photostimulation**
Although PreBötC-induced apnea disrupted memory performance,
this was one extreme case of controlling central respiratory activity.
The frequency of respiration is an essential component for coupling
neural oscillations that affect cognitive function[10]. To identify further
effects of central respiratory activity, we utilized optogenetic manip-
ulation to produce irregular rhythms and different frequencies (10 Hz
vs. 4 Hz) of PreBötC-induced activity in the conditioning paradigm.
Again, in the awake state during flat photostimulation, the frequency
of whole-body plethysmographic signals was markedly decreased by

90.5–91.1% in Cre$^+$ mice (0.61 Hz; see Fig. 6a, b). Furthermore, photo-
stimulation with 10 Hz (12.5-ms pulse-on and 87.5-ms pulse-off, duty
cycle: 12.5%) slightly decreased the frequency in Cre$^+$ mice (5.80 Hz;
$F(2, 14) = 6.61$, $P = 0.0095$; 10-Hz photostim.: $P = 0.0036$ compared to
prestimulation, post hoc pairwise $t$ test with two-sided and Bonferroni
correction) but did not change the frequency in Cre$^-$ mice ($F(2,
18) = 0.30$, $P = 0.7$, Cre$^-$ vs. Cre$^+$ in 10-Hz photostim.: $t(10.41) = 2.25$,
$P = 0.047$, Welch's $t$ test, Fig. 6c). Moreover, the variability in the
respiratory activity was quantified by the coefficient of variation (CV)
of the cycle duration of whole-body plethysmographic signals (see
"Methods"). The results showed that 10-Hz photostimulation
increased CV of the cycle duration by 14.6–15.4% in Cre$^+$ mice com-
pared to that in Cre$^-$ mice (animal type: $F(1, 16) = 3.13$, $P = 0.096$, per-
iod: $F(1, 16) = 15.10$, $P = 0.0013$; interaction: $F(1, 16) = 22.81$, $P = 0.0002$,
two-way mixed-design ANOVA; prestimulation vs. 10-Hz photostim.:
Cre$^-$: $t(9) = 0.60$, $P = 0.6$; Cre$^+$: $t(7) = 4.36$, $P = 0.003$, two-tailed paired $t$
test; Cre$^-$ vs. Cre$^+$ in 10-Hz photostim.: $t(13.72) = 3.03$, $P = 0.0092$, two-
tailed Welch's $t$ test, Fig. 6d). In addition, in Cre$^+$ mice, the pressure
amplitude of whole-body plethysmographic signals was decreased
from the prestimulation to 10-Hz photostimulation periods (Cre$^-$ vs.
Cre$^+$: $t(13.20) = 6.91$, $P = 9.9 \times 10^{-6}$, two-tailed Welch's $t$ test; Cre$^-$:
$t(9) = 0.13$, $P = 0.9$; Cre$^+$: $t(7) = 8.37$, $P = 0.00007$, two-tailed one-sample
$t$ test against zero, Fig. 6e).

In line with our conditioning paradigm, Cre$^-$ and Cre$^+$ mice
received 10-Hz photostimulation twice for 4 s after the presentation of
high tones, and the first photostimulation covered the timing of a 2-s

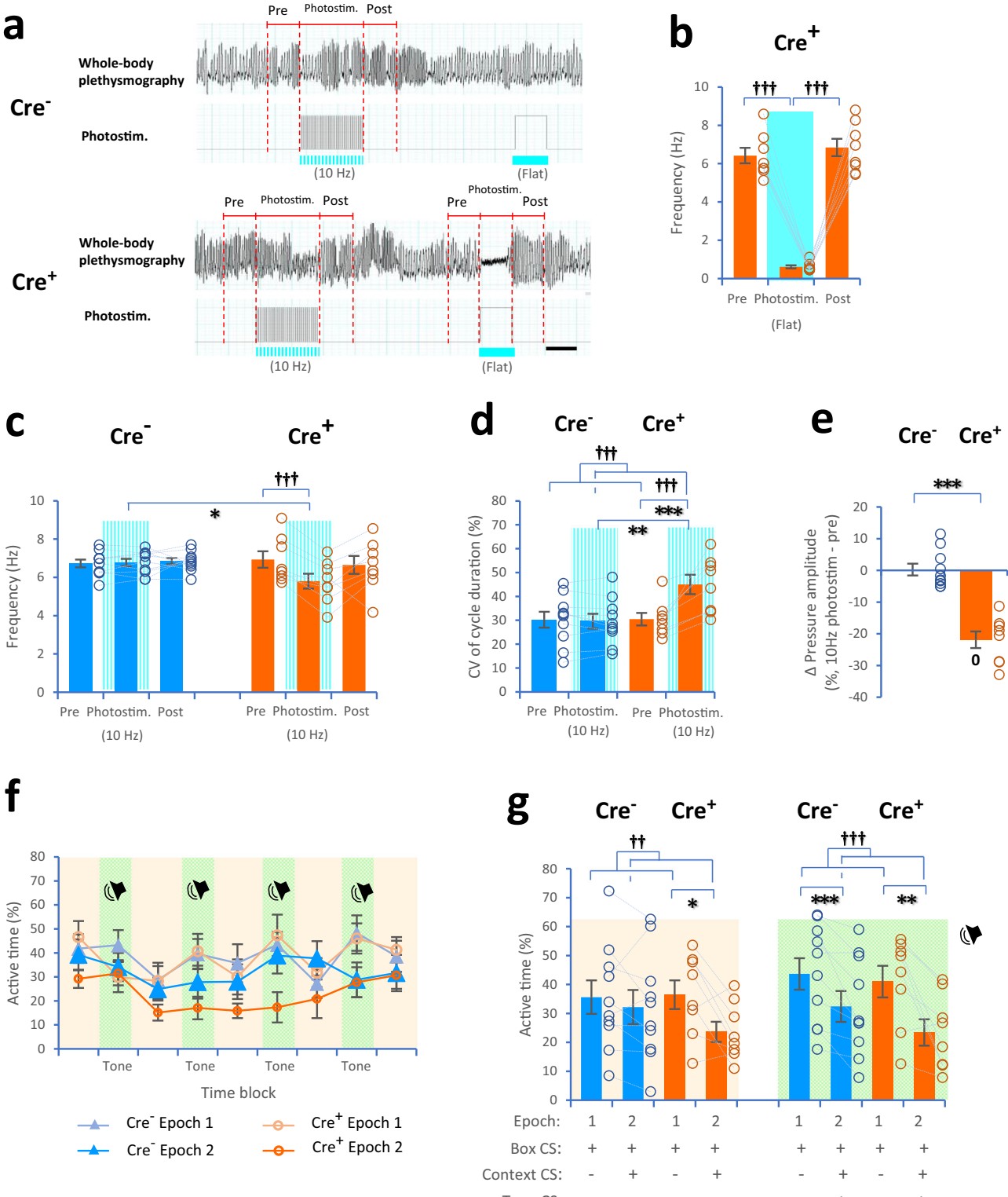

footshock during the sample sessions. Activity levels were much higher during the footshock than during the prestimulation period, and no difference was observed between Cre⁻ and Cre⁺ mice (see Supplementary Fig. 12a, b). Then, the discrimination ability of Cre⁻ and Cre⁺ mice between CS⁺ and CS⁻, which were combined with contexts and tones, was tested during the recognition session. As expected, in Cre⁻ mice, the presentation of the context/tone-dependent CS⁺ resulted in a lower active time than the presentation of context/tone-dependent CS⁻ during the onset time blocks. Interestingly, in Cre⁺

mice, during both the offset and onset time blocks, the active time was lower during CS⁺ presentation than during CS⁻ presentation (offset: animal type: $F_{(1, 16)} = 0.29$, $P = 0.6$; epoch: $F_{(1, 16)} = 7.96$, $P = 0.012$; interaction: $F_{(1, 16)} = 2.99$, $P > 0.1$; CS⁻ vs. CS⁺: Cre⁻: $t_{(9)} = 1.10$, $P = 0.3$; Cre⁺: $t_{(7)} = 2.75$, $P = 0.03$; onset: animal type: $F_{(1, 16)} = 0.70$, $P = 0.4$, epoch: $F_{(1, 16)} = 24.62$, $P = 0.00014$; interaction: $F_{(1, 16)} = 1.24$, $P = 0.3$; CS⁻ vs. CS⁺: Cre⁻: $t_{(9)} = 3.81$, $P = 0.0042$; Cre⁺: $t_{(7)} = 3.37$, $P = 0.012$, Fig. 6f, g). Our results indicated that the irregular rhythmic patterns of PreBötC-induced activity during encoding enhanced memory

**Fig. 6 | Fear conditioning with optogenetic manipulation and 10-Hz photo-stimulation. a** Plots showing the frequency of whole-body plethysmographic signals during the prestimulation (2 s), 10-Hz photostimulation (12.5-ms pulse-on and 87.5-ms pulse-off, duty cycle: 12.5%; 40 times for 4 s), flat photostimulation (2 s), and poststimulation (2 s) periods in awake Cre⁻ and Cre⁺ mice. **b** Plots showing the frequency of whole-body plethysmographic signals during the prestimulation (2 s), photostimulation (flat, 2 s), and poststimulation (2 s) periods in awake Cre⁺ mice ($P = 0.013$, Mauchly tests for sphericity; $F_{(2, 14)} = 179.0$, $P = 1.1 \times 10^{-10}$, one-way repeated-measure ANOVA with Greenhouse–Geisser correction; $P = 6.6 \times 10^{-6}$ compared to prestimulation, $P = 8.9 \times 10^{-6}$ compared to post, post hoc pairwise $t$ test with two-sided and Bonferroni correction). **c** Plots showing the frequency during the prestimulation (2 s), 10-Hz photostimulation (4 s), and poststimulation (2 s) periods in Cre⁻ (blue) and Cre⁺ (orange) mice. **d** Plots showing the variability (coefficient of variation, CV) in the cycle duration in Cre⁻ and Cre⁺ mice. **e** Plots showing changes in the pressure amplitude of whole-body plethysmographic signals from the prestimulation to 10-Hz photostimulation periods in Cre⁺ mice. **f, g** Plots showing active time between CS⁻ and CS⁺ during the offset time blocks (light orange zone) and onset time blocks (light green zone) in Cre⁻ (blue) and Cre⁺ (orange) mice. The scale bar is 2 s in (**a**). Cre⁻ mice ($n = 10$, 8 females and 2 males, blue) and Cre⁺ mice ($n = 8$, 5 females and 3 males, orange) in (**b**–**e**, **g**). †$P < 0.05$, ††$P \le 0.01$, †††$P < 0.005$ (main effects and interactions using ANOVA and post hoc pairwise comparisons), *$P < 0.05$, **$P \le 0.01$, ***$P < 0.005$ (two-tailed Welch's $t$ test and two-tailed paired $t$ test), ⁰$P < 0.05$ (two-tailed one-sample $t$ test against zero). Bar and line plots indicate means ± SEM. Circles in the graph represent individuals.

performance during the whole CS⁺ presentation (i.e., with and without tones).

## Fear conditioning using optogenetic manipulation and 4-Hz photostimulation

In the conditioning experiment with photostimulation at 4 Hz, the frequency of whole-body plethysmographic signals in Cre⁺ mice was decreased as 0.51 Hz (median) during flat photostimulation (see Fig. 7a, b), and 3.35 Hz (mean) during photostimulation at 4 Hz (162.5-ms pulse-on and 87.5-ms pulse-off, duty cycle: 65.0%; Cre⁺: $P = 0.038$, Mauchly tests for sphericity; $F_{(2, 20)} = 331.1$, $P = 4.7 \times 10^{-16}$ with Greenhouse–Geisser correction; 4-Hz photostim.: $P = 3.2 \times 10^{-10}$ compared to prestimulation, $P = 2.0 \times 10^{-10}$ compared to poststimulation, post hoc pairwise $t$ test with two-sided and Bonferroni correction; Cre⁻: $F_{(2, 16)} = 0.42$, $P = 0.7$; Cre⁻ vs. Cre⁺ in 4-Hz photostim: $t_{(10.53)} = 15.96$, $P = 1.4 \times 10^{-8}$, two-tailed Welch's $t$ test, Fig. 7c). Furthermore, 4-Hz photostimulation did not change the variability in the cycle duration in Cre⁻ and Cre⁺ mice (animal type: $F_{(1, 18)} = 2.14$, $P = 0.2$; period: $F_{(1, 18)} = 1.81$, $P = 0.2$; interaction: $F_{(1, 18)} = 0.03$, $P = 0.9$, two-way mixed-design ANOVA, Fig. 7d). Although changes in the pressure amplitude of whole-body plethysmographic signals between the prestimulation and 4-Hz photostimulation periods were below zero in both Cre⁻ and Cre⁺ mice, Cre⁺ mice had lower percentage changes than Cre⁻ mice (Cre⁻ vs. Cre⁺: $t_{(14.61)} = 5.21$, $P = 0.0001$, two-tailed Welch's $t$ test; Cre⁻: $t_{(8)} = 2.51$, $P = 0.036$; Cre⁺: $t_{(10)} = 7.12$, $P = 0.00003$, two-tailed one-sample $t$ test against zero, Fig. 7e).

In the conditioning task, no difference was observed in the activity levels of Cre⁻ and Cre⁺ mice during the footshock in the sample sessions (see Supplementary Fig. 12c, d). Regarding the discrimination ability of Cre⁺ mice, no difference in the active time was observed between the CS- and CS⁺ presentations during the onset time blocks (offset: animal type: $F_{(1, 18)} = 1.09$, $P = 0.3$; epoch: $F_{(1, 18)} = 5.99$, $P = 0.02$; interaction: $F_{(1, 18)} = 0.33$, $P = 0.6$; CS⁻ vs. CS⁺: Cre⁻: $t_{(8)} = 1.79$, $P = 0.11$; Cre⁺: $t_{(10)} = 1.68$, $P = 0.12$; onset: animal type: $F_{(1, 18)} = 1.56$, $P = 0.2$; epoch: $F_{(1, 18)} = 7.81$, $P = 0.012$; interaction: $F_{(1, 18)} = 4.20$, $P = 0.055$; CS⁻ vs. CS⁺: Cre⁻: $t_{(8)} = 3.16$, $P = 0.013$; Cre⁺: $t_{(10)} = 0.75$, $P = 0.5$, Fig. 7f, g). Strikingly, the active time during CS⁻ presentation was lower in Cre⁻ mice than in Cre⁺ mice (Cre⁻ vs. Cre⁺ in CS⁻: $t_{(18.00)} = 2.42$, $P = 0.03$, Fig. 7g). These results revealed that reducing the frequency of PreBötC-induced activity at the exact time of encoding caused a decline in memory performance.

## Discussion

Using optogenetic manipulation in *Vgat*-Cre⁺ and *Vgat*-Cre⁻ mice performing two memory tasks, we brought evidence that central respiratory activity during encoding dramatically changed the dynamics of CA3 cell ensembles and subsequent memory performance. Short-term PreBötC-induced apnea occurring at the exact time of encoding impaired novel object detection. The memory impairment occurred when 61% of the object exploration time was covered with

intermittent PreBötC-induced apnea during encoding. Moreover, PreBötC-induced apnea occurring during encoding did not produce CS⁺ associative memories in the conditioning task, where object and olfactory strategies were not effective. This apnea did not alter the dynamics of CA3 cell ensembles and increased the number of reactivation cells, which were activated in both CS⁻ and CS⁺, suggesting associative memory impairments. Furthermore, optogenetic manipulation for patterns of PreBötC-induced activity (e.g., irregular phase shift and decreased frequency) occurring during encoding resulted in different memory performance. These findings suggest that PreBötC-induced activity, which is equivalent to EI transition-dependent signals, may stimulate certain mechanisms for the orchestration of CA3 cell ensembles and might be involved in information alignment and/or memory formation.

This study showed that online encoding is modulated by central respiratory activity. Importantly, memory performance is improved or deteriorated by the frequency and regularity of PreBötC-induced activity during encoding. In the conditioning task, optogenetic manipulation with flat photostimulation (i.e., apnea) during encoding did not cause freezing behavior, in which Cre⁺ mice exhibited a type II error or false negative during CS⁺ presentation. Optogenetic manipulation with 10-Hz photostimulation induced irregular breathing cycles at the exact time of encoding, and then led to better memory performance. This finding could be supported by the theory of stochastic resonance[35] that the irregular signal inputs may improve hippocampal-dependent memory[36]. Moreover, respiratory activity induced by photostimulation at a lower frequency (i.e., 4 Hz) during encoding caused ambiguous behavior and reduced the ability of Cre⁺ mice to discriminate between CS⁻ and CS⁺ presentations, considering a different type of memory impairments (i.e., a type I error or false positive occurring during CS⁻ presentation). Cre⁺ mice slowed down during flat photostimulation and photostimulation at different frequencies, but their behavior looked natural and healthy afterwards. It is likely that the frequency and patterns of central respiratory activity might be key drivers in modulating momentary links between information sequences on the order of 100 milliseconds. The manipulation of PreBötC-induced activity may have specific effects on associative memory that could be caused by either interrupting or enhancing the corollary discharge from the PreBötC to memory structures in the hippocampus[18].

However, we cannot rule out the possibility that manipulation of PreBötC-induced apnea and degraded frequencies causes transient systemic impacts throughout the whole brain. Indeed, cardiovascular effects were observed in Cre⁺ mice, and these mice might get diminished abilities to perceive, explore, and understand what was happening, which could be similar to fleeting moments of dizziness, lightheadedness, or faintness. Thus, transient systemic impacts may affect individual Cre⁺ mice differently. More refined in vivo electrophysiological recordings from the cell ensembles and brain regions would be essential to elucidate the systemic effects or network pathways derived from the PreBötC.

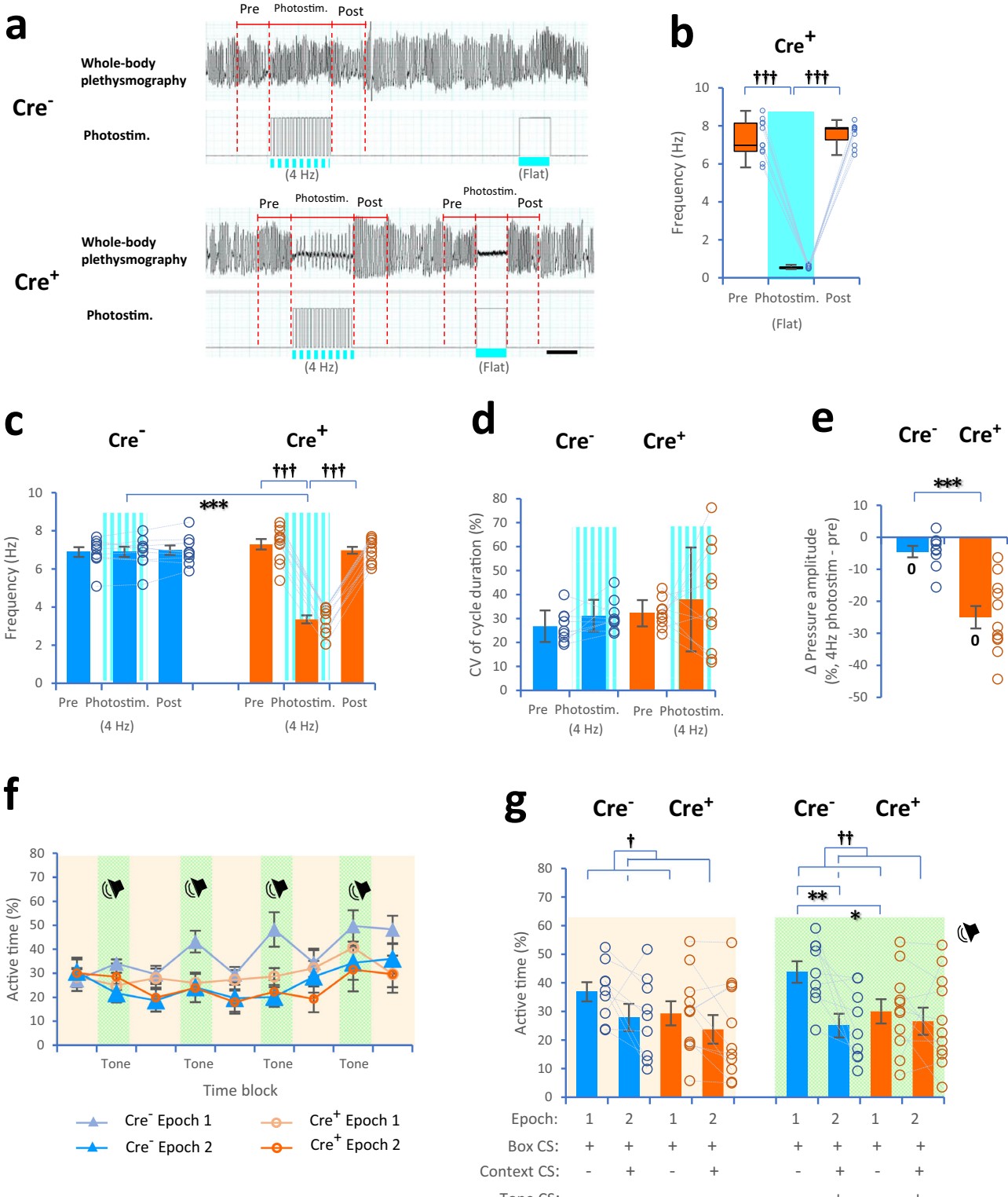

While Arc and Homer1a (H1a) proteins are crucial molecules for synapse-specific and cell-wide plasticity in the hippocampus and neocortex[37–42], the expression of *Arc*, *Fos*, and *H1a* in hippocampal and cortical neurons has been described as a "complementary genetic timer of neural activity" and applied as a coincidental marker of activation for distinct temporal events in behavioral tasks using catFISH methods[23,24,43]. *Arc* and *H1a* are considered primarily glutamatergic IEGs, whereas *Fos*, *FosB*, *Egr1* (or *zif268*), and *Npas4* are acutely induced by depolarization in

GABAergic neurons in cultures[44,45]. The catFISH methods for *Arc*, *Fos*, and *H1a* show several neural signatures during behavioral tasks. First, catFISH is used to observe anatomical features of neural activity at the cellular level and identify the cell ensemble activity per unit space in the hippocampus and prefrontal cortex. We found that cell ensemble dynamics in the distal part of CA3 (close to CA2) were recruited more than those in the proximal part of CA3 (close to the dentate gyrus), suggesting that the current conditioning paradigm might be more relevant for

**Fig. 7 | Fear conditioning with optogenetic manipulation and 4-Hz photo-stimulation. a** Plots showing the frequency of whole-body plethysmographic signals during the prestimulation (2 s), 4-Hz photostimulation (162.5-ms pulse-on and 87.5-ms pulse-off, duty cycle: 65.0%; 16 times for 4 s), flat photostimulation (2 s), and poststimulation (2 s) periods in awake Cre⁻ and Cre⁺ mice. **b** Plots showing the frequency of whole-body plethysmographic signals during the prestimulation (2 s), photostimulation (flat, 2 s), and poststimulation (2 s) periods in awake Cre⁺ mice ($\chi^2(2) = 16.91$, $P = 0.0002$, Friedman test; photostimulation: $P = 0.003$ compared to prestimulation, $P = 0.003$ compared to poststimulation, post hoc pairwise Wilcoxon signed-rank test with two-sided and Bonferroni correction). **c** Plots showing the frequency during the prestimulation (2 s), 4-Hz photostimulation (4 s), and poststimulation (2 s) periods in Cre⁻ (blue) and Cre⁺ (orange) mice. **d** Plots showing

the variability in the cycle duration in Cre⁻ and Cre⁺ mice. **e** Plots showing changes in the pressure amplitude in Cre⁺ mice. **f**, **g** Plots showing active time between CS⁻ and CS⁺ during the offset time blocks (light orange zone) and onset time blocks (light green zone) in Cre⁻ (blue) and Cre⁺ (orange) mice. The scale bar is 2 s in (**a**). Cre⁻ mice ($n = 9$, 4 females and 5 males, blue) and Cre⁺ mice ($n = 11$, 7 females and 4 males, orange) in (**b**–**e**, **g**). †$P < 0.05$, ††$P \le 0.01$, †††$P < 0.005$ (main effects and interactions using ANOVA, Friedman test, and post hoc pairwise comparisons), *$P < 0.05$, **$P \le 0.01$, ***$P < 0.005$ (two-tailed Welch's $t$ test and two-tailed paired $t$ test), $^0P < 0.05$ (two-tailed one-sample $t$ test against zero). Box plots indicate median, first, and third quantiles, and minimum and maximum values. Bar and line plots indicate means ± SEM. Circles in the graph represent individuals.

assessing contextual effects[25,34,46,47]. Second, catFISH is applied to evaluate temporal features related to the dynamics of cell ensembles in two distinct events (i.e., Epochs 1 and 2), which were approximately 20 min apart, although neural activity was detected at only two time points. Third, catFISH indicates spatio-temporal aspects of the dynamic changes between two distinct events at the behavioral and cellular levels in multiple brain areas. Our results showed that freezing behavior had opposite correlations with *Arc*-expressing cell ensembles in the hippocampus and medial prefrontal cortex. Fourth, catFISH can compensate for these features and increase the robustness using different sets of IEGs. We used two sets of catFISH (i.e., the detection of cyto-plasmic *Arc* vs. nuclear *Arc*, and the detection of nuclear *H1a* vs. nuclear *Fos*), demonstrating the similar dynamics of cell ensembles in Cre⁻ and Cre⁺ mice. Furthermore, although not directly tested, catFISH methods might detect molecular features of states of active and inactive synapses of cell ensembles that previously experienced strong activation. Arc protein is rapidly increased by strong synaptic activity and conversely accumulates in inactive synapses[48]. According to the phenomenon of inverse synaptic tagging[48], we assume that animals that previously experienced strong stimuli (i.e., CS⁺ presentations) may have less accumulation of Arc protein at active synapses in hippocampal cells and more accumulation at inactive synapses, which is reflected by the decreased *Arc* expression during CS⁻ presentation in the recognition session. These catFISH results allowed us to assess the dynamics of cell ensembles in the hippocampus.

The current optogenetic manipulation successfully activated inhibitory neurons in the PreBötC[22] and caused short-term respiratory arrest and different rhythmic patterns of PreBötC-induced activity. The phases of respiration are determined by firing patterns and the coordination of neuronal subpopulations in the brainstem and are divided into distinct phases (i.e., inspiration, postinspiration, and active expiration)[2–4]. The inspiratory-to-expiratory phase (IE) transition (the onset time of postinspiration) is considered to occur in a "gradual or convergent manner", since the IE transition is caused by a gate control mechanism of neural excitability[3] and an inspiratory off-switch mechanism of neural networks in the medulla and pons[1]. Although the parafacial respiratory group in the ventrolateral medulla is known as the expiratory rhythm generator, these neurons appear to be quiescent in mature intact rodents and generate late-expiratory bursts conditionally[5]. In contrast, the subpopulation of PreBötC excitatory neurons generates the EI transition (the onset time of inspiration) and persists bursting intrinsically, even when the PreBötC is isolated in a slice from the neonatal rodent medulla in vitro[3]. Thus, the PreBötC has an independent isolated property and EI transition-dependent signals can be determined in a divergent manner that impacts the higher brain.

The nasal respiration system is associated with the coupling of oscillatory activity in the brain. Specifically, nasal respiration entrains delta (0.5–4 Hz) oscillations in the olfactory bulb and barrel cortex,

and theta (4–8 Hz) and gamma (30–80 Hz) oscillations in the pre-frontal cortex and hippocampus[9,49–51]. During exploratory behavior and immobility in awake mice, respiration-delta/theta coupling is also recruited in these regions[52,53]. Strikingly, the disappearance of respiration-delta/theta coupling diminished the occurrence of gamma oscillations[9], suggesting that respiration-coupled delta/theta oscillations is essential for regulating gamma oscillation-related cognitive function[10]. However, whether the neural mechanisms of central respiratory activity coordinate memory traces remains to be determined. There is an alternative source of this modulation: the PreBötC can fine-tune the activity of the locus coeruleus[54], which has afferent projections preferentially to the CA3 of the hippocampus[55]. Further work is necessary to investigate the functional coordination among the CA3, locus coeruleus, and PreBötC during online brain states. These findings improve our understanding of interactions between breathing and cognitive function that underlie the mechanisms for successful performance in daily life.

## Methods
### Animals
All animal procedures were performed in accordance with the Guidelines for Proper Conduct of Animal Experiments, Science Council of Japan, and the regulations for animal experimentation of the Hyogo Medical University and were approved by the Animal Experiment Committee and the Ethical Committee at the Hyogo Medical University (18-039, 19-005, 20-049, 218001, HCM-0921). Adult *Vgat*-Cre⁺ mice (Cre⁺, males and females, B6J.129S6(FVB)-Slc32a1<tm2(cre)Low1>-MwarJ, JAX Mice 016962 B6J background, 028862, Jackson Laboratory)[56] and adult wild-type C57BL/6J mice (*Vgat*-Cre⁻ or Cre⁻, males and females, Japan Charles River) were housed with food and water available ad libitum under a 12-h light cycle in a temperature-controlled room (23 ± 1 °C and 50 ± 1% humidity) at a minimum of 90% of normal body weight. All efforts were made to minimize the number of animals used and their suffering.

### Viral vectors
*pAAV-Ef1α-DIO-hChR2(H134R)-EYFP* (provided by K. Deisseroth, Stanford University)[57] and *pAAV-Ef1α-DIO-eNpHR3.0-EYFP* (provided by K. Deisseroth, Stanford University)[58] were purchased (Addgene). AAV vectors were produced by using the AAV Helper Free Expression System (Cell Biolabs, Inc., San Diego, CA)[59].

### Microinjection
AAV injections were performed when Cre⁻ and Cre⁺ mice were 12–14 weeks old. Animals were anesthetized with isoflurane (3.0% for induction and 2.0–2.5% for maintenance, wt/vol) and placed in a ste-reotaxic apparatus (Narishige Scientific Instrument) with bregma and lambda skull landmark levels. According to the experimental protocols of Sherman and colleagues[22], two holes were drilled in the skull at predefined coordinates relative to bregma to allow the insertion of glass pipettes containing a mixture of two virus solutions (ChR2:

0.9–1.7 × 1013 genome copies per ml; eNpHR: 1.4 × 1013 genome copies per ml), which were held by a stereotaxic micromanipulator (SMM-100, Narishige) and connected to a Hamilton syringe with a pressure injection system (Motorized Stereotaxic Microinjector, IMS-20, Narishige). Each animal had bilateral injections targeted precisely to a location between the PreBötC and BötC (6.70 mm caudal to bregma, 1.25 mm lateral to the midline, and 4.65 mm ventral from the dorsal surface of the brain) according to the mouse brain atlas[60]. Once situated, 500 nl per side was slowly injected (100 nl/min) and the pipette was left in place for at least 5 min after injection to minimize backflow. The wound was closed with instant adhesives. AAV injections were performed by an investigator blinded to the animals.

### Implantation of fiber optics
One to 4 days after the injection, animals were again anesthetized with isoflurane (3.0% for induction and 2.0–2.5% for maintenance, wt/vol) and placed in a stereotaxic apparatus (Narishige). Fiber-optic cannulae with two 1.25-mm stainless steel ferrules (600 µm fiber, 2.5-mm interval, Doric Lenses), held by a stereotaxic micromanipulator (Narishige), were implanted bilaterally to a depth 350–450 µm dorsal to the PreBötC/BötC, based on pre-determined coordinates. The cannulae were glued to the skull and the wound was sealed by applying dental acrylic resins (Super-Bond; Sun Medical, Japan). Animals were returned to their home cage and allowed 2–3 weeks to recover and obtain sufficient levels of protein expression. Animals that breathed spontaneously and normally, were handled a week before the experiment.

### Photostimulation
Branching optical fibers (600-µm fiber, Doric Lenses) with 2.5-mm stainless steel ferrules were connected to the implanted cannulae via plastic sleeves. The back end of the fiber was connected to a 465-nm/595-nm (blue/yellow lights) dual wavelength laser (LEDRV_2CH_1000, Doric Lenses) via an optical rotary joint (LEDFRJ-B/A-FC, Doric Lenses). Photostimulation pulses were controlled online using LabChart software (LabChart 8.1, AD Instruments).

### Physiological apparatus, stimulation apparatus, and respiratory data
Respiratory waveforms were continuously recorded by a spirometer (AD Instruments) using an isoflurane anesthesia system (anesthesia unit, Univentor 400) and a mass flow controller (Model 8300, KOFLOC, Japan) via air variability in the closed transparent acrylic box, which was equipped with whole-body plethysmography systems[61–63]. Electrocardiogram (ECG) was continuously recorded via lead II (voltage between the left and right arm electrodes) by a differential biological amplifier (Bioamp, AD Instruments). Footshocks (2 s, 0.8 mA) were generated by a handmade copper grid inside the acrylic box with a stimulus isolator (AD Instruments) and controlled online using LabChart software (LabChart 8.1, AD Instruments). All electrical signals were sampled at 1 kHz using a PowerLab data acquisition system (PowerLab, AD Instruments) and computed online using LabChart software. The animal's exploratory behavior in the open field and acrylic box was captured by an infrared video camera (1920 × 1080 resolution, 60 Hz frames per second, HC-W870M, Panasonic, Japan) placed on the ceiling (a 60-cm distance between the camera lens and the animal).

The variability in the animal's respiratory activity in the acrylic box was calculated by the coefficient of variation (CV) of the cycle duration of the whole-body plethysmographic signals. The CV was defined as the ratio of the standard deviation to the mean. Although whole-body plethysmographic signals included small physical movement signals in awake rodents, their frequency and cycle duration almost certainly reflected the frequency and cycle duration of respiration[62].

### Object-recognition memory and optogenetic manipulation
A spontaneous object version of the recognition memory task[29] was employed with some modifications. The apparatus was an open field inside a brown-colored chamber (interior: width × length × height = 27.5 × 39.5 × 55.0 cm$^3$, made of polyvinyl chloride), with extra maze cues available in the experimental room. Two identical objects with column shapes made of pottery and metal were used.

Each animal was habituated to the open field over 4 days. Each animal explored the empty open field for 6 min twice during days 1 and 2. Subsequently, on days 3 and 4, animals were habituated to the experimental conditions of the testing day (two 6-min trials with a 20-min intertrial interval) and the detection of two novel objects placed 6 cm from the walls in the open field, which were not used during the testing day.

On the testing day (day 5), animals performed the behavioral procedure, including a sample epoch (6 min), a delay (20 min), and a recognition epoch (6 min, Fig. 2a). During the sample epoch, animals were exposed to two identical objects (i.e., pottery). During the sample epoch, the PreBötC was exposed to manual photostimulation for 2 s by an investigator when the animals were within a 2-cm distance of each object. The animals were returned to their home cages for 20 min after the sample epoch. Then, two objects were placed in the open field at exactly the same locations as those used during the sample epoch. One object was one of the same objects used during the sample epoch (i.e., pottery), and the other was a novel object that had not been encountered previously (i.e., metal). The original objects were replaced by the novel object in a pseudorandom manner. After each epoch, the open field and the objects were cleaned with 70% ethanol and distilled water.

### Data analysis for object-recognition memory
Successful memory performance was assessed according to the natural preference of rodents to spend more time exploring novel objects than familiar objects[28]. The object exploration time was defined as the time directing their nose within a 2 cm distance of the object. The exploratory behavior of the animals was scored with the standard discrimination ratio d2 (d2 = exploration time for the novel object exploration time for the familiar object/ exploration time for both the novel and familiar objects)[28]. Thus, the discrimination ratio versus zero significantly reflects successful memory performance. The exploratory behavior during the sample epoch was scored as the discrimination ratio for the right-side object, which was replaced by a novel object during the recognition epoch. The object-recognition memory data were collected from distinct animals.

To measure the animal exploration of these objects in the open field, image series were preprocessed using ImageJ software (ImageJ 1.51k, NIH, USA; http://imagej.nih.gov/ij) and MATLAB algorithms with the computer vision toolbox and image processing toolbox (MATLAB R2018b, MathWorks, USA, http://www.mathworks.com). The image series acquired in the open field (275 × 395 mm$^2$) during each epoch were converted from 60 to 20 Hz frame sizes with consistent grayscale thresholds. Temporal series of the pixel sizes were acquired when the animal reached a position within 2 cm of each object in the open field. The object exploration time data were automatically collected without a biased criterion of judgment.

### Preliminary experiments with the fear-conditioning paradigm
We employed a fear-conditioning paradigm that was designed with a pair of conditioned stimuli (CS$^+$) and the conditioned inhibition of fear (CS$^-$), as described in ref. 33, with some modifications. As animals naturally prefer to spend time in the dark rather than in a bright place in passive (inhibitory) avoidance[64–66], we used chambers with black and white walls for both the CS$^-$ and CS$^+$.

The conditioning paradigm consisted of two 5-min epochs (Epochs 1 and 2) on 2 to 4 consecutive days for the sample sessions,

and 24 h later the animals' ability to discriminate between the CS⁻ and CS⁺ was evaluated during the recognition session (Supplementary Figs. 4–8). Each animal was placed into a closed transparent acrylic box. In the box, a handmade copper grid was placed on the floor (interior: width × length × height = $5.2 × 8.1 × 13.0$ cm³), and footshocks (2 s, 0.8 mA) were delivered (Supplementary Fig. 4a). The acrylic box was placed away from either black round walls (interior: short axis × long axis × height = $27.7 × 49.7 × 55.0$ cm³) or white flat walls (interior: width × length × height = $27.7 × 49.7 × 55.0$ cm³). Individual animals were divided into the following groups: (i) the animals in Groups A–D are habituated to an acrylic box for 2 days, and then performed the fear-conditioning task for 4 days (3 for the sample session and 1 for the recognition session, see Supplementary Fig. 5a, c, e, g); (ii) the animals in Groups E and F are habituated to the box for 2 days, and performed the task for 5 days (4 for the sample session and 1 for the recognition session, Supplementary Fig. 7a, b); and (iii) the animals in Group G are habituated to the box for 2 days, and performed the task for 3 days (2 for the sample session and 1 for the recognition session, Supplementary Fig. 8a).

On day 1 of the sample sessions, each animal in Group A was placed in the acrylic box, which was surrounded by white walls (Epoch 1). After a few 10-s periods of free exploration by the animals, the chamber was brightened (1400 lx illuminance) during an onset time block for 20 s (Supplementary Fig. 4b), and an unsignaled footshock with a trace interval of 2–3 s was delivered after each onset time block. Then, offset time blocks (5 lx illuminance) with 30- to 50-s intervals and onset time blocks with footshocks were alternately repeated four times in 5 min (Supplementary Fig. 4c). Twenty minutes after being returned to its home cage, the animal was placed in the same acrylic box, which was surrounded by different walls (i.e., black walls, Epoch 2). The chamber was dark (0 lx illuminance) during the onset time block and the animal did not receive a footshock. Then, onset time blocks without footshocks were repeated four times. The animals in Group A underwent conditioning on days 2 and 3 in consideration of the order effect (Supplementary Fig. 5a, b). The animals in Group B underwent counterpart conditioning (Supplementary Fig. 5c, d). During the recognition session (day 4), we evaluated the animals' ability to distinguish between distal spatial contexts (i.e., black walls with darkness vs. white walls with brightness) under the same proximal cue (i.e., the acrylic box) based on the association with either the CS⁻ or CS⁺ (Supplementary Fig. 6a). The temporal activity parameters in the acrylic box (i.e., active time) were estimated during each onset and offset time block (see below) instead of the "conventional freezing index", which is defined as a freezing state unless the animals move for more than 2 s.

The animals in Groups C and D performed a fear-conditioning task following similar paradigms as those previously described for the animals in Groups A and B, respectively, with additional auditory stimuli, which were delivered during the onset time blocks (i.e., black walls with 0 lx illuminance and low tones; white walls with 1400 lx illuminance and high tones). On day 1 of the sample sessions, each animal in Group C was placed in the acrylic box, which was surrounded by white walls (Supplementary Fig. 5e, f). The chamber was brightened (1400 lx illuminance), and high tones were presented (400 Hz, 80 dB, with 55 dB of background noise) during an onset time block, and a footshock was delivered. Twenty minutes after being returned to its home cage, the animal was again placed in the acrylic box, which was surrounded by a different chamber (i.e., black walls). The chamber became dark (0 lx illuminance), and low tones were presented (166 Hz, 80 dB, with 55 dB of background noise) during the onset time block, and the animal did not receive a footshock. The animals in Group D performed counterpart conditioning (Supplementary Fig. 5g, h). During the recognition session (day 4), the animals were tested to distinguish between mixtures of distal spatial contexts and auditory stimuli

(i.e., black walls with darkness and low tones vs. white walls with brightness and high tones) under the same proximal cue (i.e., the acrylic box) in association with either the CS⁻ or CS⁺ (Supplementary Fig. 6c).

The animals in Groups E and F performed the task following similar paradigms as those previously described for the animals in Groups A and B, respectively, with an additional sample session (Supplementary Fig. 7a–c). In Epoch 1 of the additional sample session on day 4, each animal in Group E was placed in a novel acrylic box with a floor wire grid, and the box was surrounded by black walls. The chamber became dark during the onset time block and the animal did not receive a footshock. Twenty minutes later, the animal was placed in the same acrylic box surrounded by a different chamber (i.e., white walls). The chamber was brightened during the onset time block, and a footshock was delivered. The animals in Group F were performed counterpart conditioning. During the recognition session, the ability of the animals to distinguish spatial contexts with different boxes based on the association with the CS⁻ or CS⁺ was evaluated (Supplementary Fig. 7d).

The animals in Group G performed the task following a similar paradigm as that performed by the animals in Group C with modifications, including less sample sessions (2-day sample sessions) and reduced footshocks (four footshocks per day) (Supplementary Fig. 8a, b). Moreover, bright light (i.e., 1400 l× illuminance) was maintained during the sample and recognition sessions. During the recognition session, the animals were tested to distinguish between the CS- and CS+ in the modified task (Supplementary Fig. 8c).

### Fear conditioning using optogenetic manipulation

Cre⁻ and Cre⁺ mice performed the task following the Group G paradigm (Supplementary Fig. 8 and Fig. 3). The animal's head was connected to optic fiber cables on the celling (Fig. 3a). The acrylic box lid was removed because animals' movements are not restricted by the cables in the box. Cre⁻ and Cre⁺ mice received photostimulation twice for 4 s, the first of which covered the timing of a foootshock, for a total of 8 photostimulation periods during each sample session (Fig. 3b–e). In the current study, we utilized three photostimulation sequences: flat photostimulation (Fig. 3), 10-Hz photostimulation (12.5-ms pulse-on and 87.5-ms pulse-off, duty cycle: 12.5%, 40 times for 4 s, Fig. 6), and 4-Hz photostimulation (162.5-ms pulse-on and 87.5-ms pulse-off, duty cycle: 65.0%, 16 times for 4 s, Fig. 7). Data for fear conditioning using optogenetic manipulation were acquired from distinct animals.

### Data analysis for fear conditioning

During the recognition session of the fear-conditioning experiment, we measured 9i) the behavioral parameters in each time block—the temporal duration of animal exploration in the acrylic box—with an infrared video camera, and (ii) respiratory parameters—each respiratory duration during whole-body plethysmography—using Powerlab with a spirometer (only in the case of the closed acrylic box experiments).

Image series of the animal exploratory behavior were processed using ImageJ software (ImageJ 1.51k, NIH) and MATLAB algorithms with the computer vision toolbox and image processing toolbox (MATLAB R2018b, MathWorks). The image series in the acrylic box during the offset and onset time blocks were converted from 60 Hz to 20 Hz frames and processed with consistent thresholds of grayscale, brightness and contrast, resulting in a series of black and white images. Adjacent windows in every 50-ms frame of the image series were repeatedly subtracted between the current image and its one-back image, and the number of nonmatching pixels in each frame pair was summed to represent the "activity levels". Furthermore, the temporal series of the number of nonmatching pixels was subtracted from the baseline number of the image-background level

as the "active time" in each time block during CS⁻ and CS⁺ presentation.

## Brain collection
Immediately after Epoch 2 in the fear-conditioning experiment, the animals were deeply anesthetized with isoflurane, decapitated, and immediately cooled on ice. The brains were then collected, frozen in powdered dry ice, and subsequently stored at −80 °C. The brains were coronally sectioned on a cryostat (10 μm thickness, Cryostar NX50, Thermo Fisher Scientific), collected on polylysine-coated slides, and stored at −80 °C.

## RNA probe preparation
DNA templates were generated by RT-PCR. The Arc DNA template was designed to amplify a fragment of the mouse Arc gene from bases 1699–2851 (NCBI RefSeq: NC_000081.7, provided by T. Kitsukawa, Ritsumeikan University, see Supplementary Fig. 9). The Fos DNA template was designed to amplify a fragment containing an intron sequence of the mouse Fos gene from bases 292–1061 (NC_000078.7, provided by T. Kitsukawa, Ritsumeikan University). The Homer1a (H1a) DNA template was designed to amplify a fragment on the 3′-untranslated region of transcript variant S of the mouse H1a gene from bases 52,325–53,088 (NC_000079.7, provided by K. Akama, Rockefeller University). Antisense RNA probes were synthesized with a mixture of either digoxigenin-labeled UTP (Roche Diagnostics) or fluorescein-labeled UTP (Roche Diagnostics) and purified using Probe quant G-50 Micro columns (GE Healthcare).

## In situ hybridization histochemistry
Regarding fluorescent in situ hybridization histochemistry[25,26,67], sections were fixed with 4% paraformaldehyde in sterile 0.1 M PBS. Sections were rinsed in PBS and acetylated with 0.25% acetic anhydride in 0.1 M triethanolamine-HCl. Following prehybridization incubation, hybridization solution (50% formamide, 5 × SSC, Denhardt's solution (2.5×), 250 μg/ml yeast tRNA, 500 μg/ml denatured salmon sperm DNA, 0.2 ng/μl digoxigenin-labeled Arc probe, 0.2 ng/μl digoxigenin-labeled or fluorescein-labeled Fos probe, and 0.2 ng/μl digoxigenin-labeled H1a probe) was applied to each slide (200 μl). The sections were coverslipped and incubated in a humidified environment at 65 °C for 18 h. The sections were then rinsed in 5 × SSC and 0.2 × SSC at 65 °C for 1 h. The sections were incubated with 3% bovine serum albumin in TBST buffer (0.1 M Tris−HCl pH 7.4, 0.15 M NaCl, 0.05% Tween 20) at room temperature, and then incubated with anti-digoxigenin-POD antibody (1/1000 dilution, Fab fragments from sheep, 11207733910, Roche Diagnostics) in BSA/TBST at room temperature for 1 h. The sections were rinsed in TBST, and the signal was amplified using the Tyramide Signal Amplification Cy5 system (TSA Cy5 plus system, Akoya Biosciences). For double labeling, the sections were incubated with 1% H₂O₂ in TBST and with BSA/TBST and then incubated with anti-fluorescein-POD antibody (1/500 dilution, Fab fragments from sheep, 11426346910, Roche Diagnostics) at room temperature for 1 h. The sections were rinsed in TBST, and the signal was amplified using the Tyramide Signal Amplification Cy3 system (TSA Cy3 plus system, Akoya Biosciences). Then, the sections were counterstained with DAPI (1/50,000 dilution, Life Technologies), and coverslipped. As controls for the staining, detection was also performed with the Sst antisense probe, which led to interneuronal staining in the hippocampus.

## Image acquisition
Fluorescent images in the hippocampus were captured using a ×20 objective on a laser scanning confocal microscope (LSM 780, Carl Zeiss). A tiling z-stack of images was acquired for each region of interest in CA3 and CA1 of the hippocampus, anterior cingulate cortex (ACC), prelimbic/intralimbic cortex (PL/IL), and claustrum from non-adjacent sections (pixel size: $0.83 \times 0.83$ μm²; CA3: tiling size: $3 \times 3$, image size: $1.19 \times 1.19$ mm²; CA1: tiling size: $4 \times 2$, image size: $1.57 \times 0.807$ mm²; medial prefrontal cortex: tiling size: $3 \times 4$, image size: $1.19 \times 1.57$ mm²; claustrum: tiling size: $5 \times 2$, image size: $1.95 \times 0.807$ mm²). The same laser intensity, detector gain and offset, and detector digital gain settings were used to capture all confocal image. Regions of interest were chosen from the dorsal hippocampus and prefrontal cortex according to the mouse brain atlas[60]. For the investigation of cell ensembles in the hippocampus, the distal and proximal parts of CA3 and CA1 at the septal level were selected at −1.70 to −1.95 mm anteroposterior levels defined from the bregma (Fig. 4a and Supplementary Fig. 8a). Furthermore, the ACC, PL/IL, and claustrum were selected at 1.60 to 1.20 mm anteroposterior levels defined from the bregma (Fig. 4j and Supplementary Fig. 9a). Of note, the claustrum has strong afferent and effect projections to the ACC[68].

## Data analysis for the catFISH method
A series of confocal images were processed using codes on MATLAB with image processing toolbox (MATLAB R2021b, MathWorks). Regarding Arc catFISH, we estimated the proportion of cells expressing "cytoplasmic Arc-positive" and "nuclear Arc-positive" signals and cells coexpressing cytoplasmic and nuclear Arc-positive signals (i.e., reactivation of the cells that were initially activated during Epoch 1) in the hippocampus and prefrontal cortex as described[23,25,43,69] with modifications. Nonneuron-like nuclei (~5 μm in diameter) with intensely bright and uniform DAPI staining were excluded, and neuronal nuclei identified as large and diffusely DAPI-stained were included. Morphological filtering was applied to image pixels for the DAPI and Cy5 signals, with consistent criteria among all comparable images[69]. Nuclear Arc-positive signals were defined by the nuclei carrying Cy5 tags within themselves. Meanwhile, a shell zone surrounding the outside of each DAPI-stained nucleus was obtained by subtracting its morphological dilation. Then, the overlaying signals between the zone and the Cy5-staining signals were defined as cytoplasmic Arc-positive signals. The number of cells expressing cytoplasmic Arc and nuclear Arc and the number of cells coexpressing Arc-positive signals were calculated in each region of interest from at least three nonadjacent sections[70].

Moreover, we defined cells expressing "nuclear Fos-positive" signals and/or "nuclear H1a-positive" signals for DAPI-stained nuclei carrying Cy5 (red) and/or Cy3 (green) tags[26,43,69]. Then, the number of cells expressing nuclear Fos and nuclear H1a-positive signals and the number of cells coexpressing Fos and H1a-positive signals were calculated in each region of interest from at least three nonadjacent sections.

To estimate the number of cells coexpressing Arc-positive signals and the number of cells coexpressing Fos and H1a-positive signals, the real number of cells coexpressing these signals was compared with chance, which was calculated as the number of cytoplasmic Arc-positive cells x the number of nuclear Arc-positive cells, or the number of nuclear Fos-positive cells x the number of nuclear H1a-positive cells. The number of positive and negative cells was determined automatically without a biased criterion of judgment.

## Statistical analysis
We tested for normality and sphericity using the Shapiro−Wilk normality test and Mauchly test for sphericity. We used a two-tailed Welch's $t$ test, two-tailed paired $t$ test, one-way repeated-measures ANOVA, two-way mixed-design ANOVA, post hoc pairwise comparisons using a paired $t$ test with two-sided and Bonferroni correction, a two-tailed one-sample $t$ test against zero, and Pearson's product-moment correlation analysis. If the assumption of normality was violated, we performed the two-sided nonparametric Mann−Whitney $U$ test, two-sided Wilcoxon signed-rank test, and Friedman test followed by post hoc pairwise comparisons using the Wilcoxon signed-rank test with two-sided and Bonferroni correction. If the assumption of sphericity was violated, we applied the Greenhouse−Geisser

correction. All statistical analyses were performed using R version 3.6.1 software (R Core Team, R Foundation for Statistical Computing, Vienna, Austria, 2019, https://www.R-project.org/).

## Reporting summary

Further information on research design is available in the Nature Portfolio Reporting Summary linked to this article.

## Data availability

The data that support the main findings of this study are available from the corresponding author upon request. The datasets used in this study are available at https://github.com/nakamunh/Nakamura_2023_Nat_Commun. Source data are provided with this paper.

## Code availability

Custom MATLAB codes generated in this study are available at https://github.com/nakamunh/Nakamura_2023_Nat_Commun.

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

## Acknowledgements

We thank Takashi Maejima (Kanazawa University, Japan) and Kazuo Funabiki (Kyoto University, Japan) for technical supports of optogenetic experiments, Constantine Pavlides (Tsukuba University, Japan), Lukas Beichert (Tubingen University, Germany), and Toshifumi Morinaga (Yamaguchi University, Japan) for technical supports of behavioral paradigms, Keith Akama (Rockefeller University, USA), Takashi Kitsukawa (Ritsumeikan University, Japan), and Hironobu Eguchi (Hyogo Medical University, Japan) for the preparation of plasmid DNAs, Ayumi Nakamura (Hyogo Medical University) for transgenic mouse breeding, and Yasumasa Okada (Murayama Medical Center, Japan) for comments on the manuscript and respiratory function. This work was supported by grants from the Hyogo Innovative Challenge, Hyogo Medical University (N.H.N.), the Takeda Science Foundation (N.H.N.), and the Grant-in-Aid for Scientific Research (18K06533, 22K07335) of the Japan Society for the Promotion of Science (N.H.N.).

## Author contributions

N.H.N. contributed to the study design; N.H.N. contributed to the acquisition of data; N.H.N., H.F., K.K., and Y.O. contributed to methodology; N.H.N. contributed to data analysis; N.H.N. and Y.O. contributed to data interpretation; N.H.N. contributed to the drafting of the manuscript; N.H.N., H.F., K.K., and Y.O. contributed to the final manuscript.

## Competing interests

The authors declare no competing interests.
