## [Peer Review File · Nature Communications]

Hippocampal ensemble dynamics and memory performance are modulated by respiration during encoding.Reviewers' comments:

Reviewer #1 (Remarks to the Author):

In recent years, many brain regions including the hippocampus and prefrontal cortex are reported to show respiration entrained neural activity, especially in offline states (freezing, sleeping, immobile, etc). However, whether respiration entrained activity plays any role in online states such as memory encoding remains unknown. Nakamura et al. addressed this question by inducing apnea during memory encoding phase in two behavioral tasks. By optogenetically activating Vgat+ neurons in the preBotzinger complex, the authors could reliably induce apnea with temporal control. This manipulation led to impaired memory in the object recognition task as well as in the fear conditioning task. Furthermore, by examining Arc expression in mice that went through the fear conditioning task, they found altered cell ensembles in the hippocampus (CA3 and CA1 subdivisions) and the prefrontal cortex (the anterior cingulate cortex). The behavioral tests were well designed, and the results were interesting. However, the manuscript falls short in establishing a causal relationship between respiration coordinated hippocampal ensemble dynamics and memory encoding.

Major points:

1. The authors showed that apnea coincided with the encoding phase impaired memory retrieval. However, the resulted memory deficits could be due to other factors (e.g., altered internal states or attention), not necessarily due to respiration coordinated hippocampal activity as claimed in the title. To firmly establish the causality, one may need to manipulate the respiration rate during memory encoding and link with the behavioral effects – apnea is one extreme case.
2. The authors used Arc expression (cytoplasmic and nuclear) as a readout for dynamics of hippocampal cell ensembles. Although the findings are interesting, this approach does not provide the temporal resolution necessary to address the question compared to other methods such as in vivo electrophysiological recordings.
3. In Figure 4 and Extended Data Fig. 8, there was a higher percentage of Arc+ cells in Cre+ than Cre- mice, indicating that more hippocampal CA3 and CA1 cells were active. Given that Cre+ mice had apnea and reduced memory performance as shown in Fig. 3, it would be helpful if the authors could provide an interpretation and/or discuss the implications of this finding. Is it known whether these Arc+ cells are excitatory or inhibitory? Double staining with specific markers will help to reveal the identity of these neurons.

Minor points:

1. The respiration related signals in the hippocampus could arise from several distinct sources, including the brainstem breathing center and/or sensory feedback via nasal breathing. The current dataset does not prove or rule out either. The authors may want to make their statements more inclusive and

consistent. In the first paragraph of introduction, the authors mentioned nasal respiration modulates oscillatory activity in the hippocampus, while in other parts (e.g., abstract), they talked about respiratory signals equivalent to excitatory-to-inspiratory phase transition, implying a brainstem origin.

2. In the object recognition memory task (Figure 2), was the novel object always placed at the right-side? Should this be randomized to eliminate the potential side preference?
3. For parametric statistical tests, normal distribution of the datasets should be verified. Otherwise, non-parametric tests should be used.
4. The phrase “Preliminary experiments with fear conditioning without object strategies” could be simplified to enhance readability.
5. Line 104. Please define “RR interval”.
6. Figure 1. “plethysmography” is misspelled as “pletysmography”.
7. Line 129. “a higher discrimination ratio to zero”. Use “above” instead of “to”.
8. Line 243. Anterior cingulate cortex is abbreviated as Cg1. In Figure 4, it is labeled as ACC.
9. Line 259-260. Use “co-expressed” instead of “co-expression”, “were” instead of “was”.
10. Some places could benefit from rephrasing/editing. For example, sentences in Line 67-68 and Line 92-95.

Reviewer #2 (Remarks to the Author):

The authors of this study employ optogenetic manipulations in behaving mice to test the hypothesis that respiratory activity, understood as a signal that coordinates activity timing throughout the brain can contribute to temporally organizing hippocampal neuronal ensembles' activity during memory encoding.

They use two behavioral paradigms, novel object recognition and fear conditioning, to assess the memory capabilities of mice. During memory acquisition, they deploy optogenetic manipulation, which silences neurons in the Pre-Botzinger area and causes a transient state of apnea in the mice.

They find behavioral effects in both tests for the optogenetic manipulation, and further assess the dynamics of the neural response in the hippocampus using the catFISH technique, assessing the activity of the immediate-early gene Arc.

The optogenetic experiments seem to be well executed from a technical point of view. The manuscript is well-written and relatively easy to follow.

The authors conclude that “central inspiratory activity, equivalent to signals of expiratory-to-inspiratory phase transition, would constitute the dynamics of hippocampal cell ensembles during “online” memory processes”. As stated, the conclusion is a bit difficult (i.e., what does it mean that inspiratory activity “constitutes the dynamics” of hippocampal ensembles?), although I think I get the main idea.

However, I have some critical concerns regarding the interpretation of behavioral experiments. The data does not necessarily point to a memory effect caused by a respiratory signal not arriving at different brain areas. Instead, it could be a general systemic effect of the apnea, a more straightforward explanation. This gets mixed with difficulty following some of the descriptions of the experimental protocols. I explain my concerns more specifically in the points below.

Major points

1) I want to focus on the behavioral results of Fig 2b, specifically in the Cre+ group:

First, the discrimination ratio index variability is substantial during the sample epoch. Second, six of nine mice display positive discrimination index values, indicating spending more time with the new object. Third, there are significant changes in the discrimination index between sample and recognition, regardless of the direction of these changes. Fourth, a 2-second period is substantial for a mouse's number of respiratory cycles. A 4-6 Hz respiration rate inhibited for 2 seconds amounts to skipping between 8 and 12 breathing cycles.

(Statistical tests show differences, but this would happen under several scenarios, especially with this sample size. The fact of the considerable variability is my main concern.)

These three points left me wondering about the effect of photostimulation on the mice. If they got their breathing interrupted, the impact of that interruption is likely to be systemic. For example, they could get diminished abilities to perceive, explore, track their position, or understand what is generally happening. Similar to a fleeting moment of dizziness, light-headedness, and/or faintness. And this general state of confusion explains why they vary widely in their behavior. Some of them seem to discriminate, and some others do not, because a transient systemic impact would likely hit differently to different individuals. Therefore, we could have something of this sort instead of a specific effect on memory caused by interrupting the corollary discharge from PreBot to brain structures. So how can we rule out this possibility?

2) I find the description of the design for the fear conditioning experiments confusing. For example, in Figure 3b, the diagram shows that on Day 3, there is no footshock. But, conversely, Extended Data Fig.5

shows a footshock for all groups on Day 3. I might be missing something, but I couldn't understand this by following the descriptions provided in the methods section.

Also, from Figure 3, I understood that all groups received tones as part of the protocol. However, from Extended Data Fig.5, groups A and B did NOT receive tones.

The title of figure 3 reads "Fear conditioning without object effects and optogenetic manipulations", and I don't understand what the authors mean by "object effects". Also, the title seems to indicate there are no optogenetic manipulations.

In general, the rationale for the protocol and the actual sequence of the protocol were quite hard to follow for me. I think the manuscript would benefit from more precise descriptions and more explicit explanations for the design's rationale.

But even after understanding the protocol, I see the same issue I described for the object recognition task. Again, there is considerable variability across individuals for the % active time (Fig 3 h, i). Half of Cre- mice seem to decrease their active time during no-tone periods. In Cre+, half of the group reduces their active time, and the other half increments it. Variability is again quite significant for the tone period in the Cre+ mice. The wide behavioral variation suggests that the mice might be confused instead of having a specific memory problem. It is hard to decide what is going on with this sample size.

3) The dynamics and correlations found in the expression of Arc can be interpreted in several ways, given the possibility of systemic effects described earlier. The evidence provided by the expression of this IEG does not solve or address the issues raised earlier (points 1 and 2).

Minor points.

Figure 1e,1f. Please provide individual data points

Methods:

line 18. "was" should be "as"

line 142. it says, "In the recognition session (day 4)". I thought recognition sessions occurred on day 3.

Fig3. There is no mention of the groups (A, B, C, D) in the figure, which are essential for understanding the design. This information should be added to the figure.

To which group do the behavioral results belong?

Response to Reviews' comments

NCOMMS-22-07329-T

Hippocampal ensemble dynamics in memory encoding coordinated by respiration.

Corresponding Author: Nozomu H. Nakamura

We have marked the corrected words and sentences with blue in the manuscript.

Reviewer #1 (Remarks to the Author):

... However, the manuscript falls short in establishing a causal relationship between respiration coordinated hippocampal ensemble dynamics and memory encoding.

R1 Q1 (1 of 13). Major points:

1. The authors showed that apnea coincided with the encoding phase impaired memory retrieval. However, the resulted memory deficits could be due to other factors (e.g., altered internal states or attention), not necessarily due to respiration coordinated hippocampal activity as claimed in the title. To firmly establish the causality, one may need to manipulate the respiration rate during memory encoding and link with the behavioral effects – apnea is one extreme case.

R1 A1. We appreciate the great suggestion for developing the causality of this finding. As the reviewer's suggestion, we have tested an additional optogenetic manipulation during memory encoding to determine the effect of regularity of respiratory activity (i.e., respiratory rates). Text has been added to explain about the additional experiment and its results.

p.11 (Results): "Even though PreBötC-induced apnea disturbed memory performance, this was one extreme case of controlling central respiratory activity. Regularity of respiratory activity is one of essential components for coupling neural oscillations and affecting cognitive function⁹. To identify the effect of regularity of respiratory activity, we utilized optogenetic manipulation to produce irregular rhythms of respiratory activity in the conditioning paradigm."

p.12 (Results): "Then, we tested their ability to distinguish between CS⁺ and CS⁻ combined with contexts and tones. In both Cre⁻ and Cre⁺ mice during onset time blocks, presentation of the context/tone-dependent CS⁺ exhibited a lower active time than context/tone-dependent CS⁻ presentation (offset: animal-type: $F(1, 13) = 0.22$, $p > 0.1$; epoch: $F(1, 13) = 11.24$, $p = 0.005$; interaction: $F(1, 13) = 7.72$, $p = 0.02$; CS⁻ vs. CS⁺: Cre⁻: $t(8) = 0.76$, $p > 0.1$; Cre⁺: $t(5) = 5.28$, $p = 0.003$; onset: animal-type: $F(1, 13) = 0.97$, $p > 0.1$, epoch: $F(1, 13) = 16.66$, $p = 0.001$; interaction:

$F(1, 13) = 1.22, p > 0.1$; CS⁻ vs. CS⁺: Cre⁻: $t(8) = 3.40, p = 0.009$; Cre⁺: $t(5) = 2.59, p = 0.049$, Fig. 6e,f). These results showed that irregular rhythmic patterns of inspiratory activity derived from PreBötC maintained memory performance; in other words, memory performance was recovered by components of inspiratory activity, which occurred at the exact time of encoding.”

p.13 (Discussion): “Meanwhile, irregular rhythmic patterns of inspiratory activity (i.e., 12.3-14.2% CV) occurring during encoding produced a CS⁺-associated memory. However, we cannot rule out the possibility that PreBötC-induced apnea causes a transient systemic impact on the whole brain. These findings suggest that components of EI-transition-dependent signals derived from the PreBötC may stimulate a certain regulatory mechanism for the orchestration of CA3 cell ensembles and might be involved in information manipulation rather than memory storage.”

p.16 (Discussion): “Our optogenetic manipulations showed that Cre⁺ mice recovered freezing behavior in the conditioning task when irregular rhythmic patterns of inspiratory activity (12-14% CV) occurred at the exact time of encoding. Since averaged respiratory frequency in Cre⁺ mice was not altered under this manipulation (5.9-7.2 Hz), it is likely that a respiratory rate might be a clue component to modulate the coordination with cognitive function. Moreover, this manipulation can be assigned to maintain each phasic period of PreBötC activation (i.e., each inspiratory phase) because a half wavelength of respiratory cycles can be assigned as 0.083 s at 6 Hz., and an offset duration of the photostimulation was 0.0875 s (i.e., 0.1 - 0.0125 s pulse width) during the 10-Hz optogenetic manipulation. Thus, we propose that components of inspiratory activity derived from PreBötC (e.g., rates and phases) could be key factors for a certain mechanism of having momentary links between information sequences on a sub-msec temporal scale.”

R1 Q2 (2 of 13).

2. The authors used Arc expression (cytoplasmic and nuclear) as a readout for dynamics of hippocampal cell ensembles. Although the findings are interesting, this approach does not provide the temporal resolution necessary to address the question compared to other methods such as in vivo electrophysiological recordings.

R1 A2. We greatly appreciate the reviewer state. We totally agree that in vivo electrophysiological recordings can provide a much greater temporal resolution. However, the catFISH approach still has several benefits to determine dynamic changes of cell ensembles in the hippocampus and prefrontal cortex, and were used in several recent papers (e.g., Nomoto et al., 2016, nat commun 7:12319; Sakaguchi et al., 2018, nat commun 9:3526; Suzuki et al., 2022, nat commun. 13:41). Indeed, the catFISH approach can provide changes in cell ensembles per unit space of the hippocampus in two distinct events (i.e., Epochs 1 and 2). Importantly, to compensate the reliability, we have further performed the other set of IEG catFISH for nuclear *Fos* and nuclear

Homer 1a, which IEGs are also confirmed to determine the dynamics of cell ensembles (Bottai et al., 2001; Vazdarjanova and Guzowski 2004; Chawla et al., 2004; Nakamura and Sauvage 2016). We revised the description of this approach as follows:

pp.13-14 (Discussion): “The catFISH techniques for *Arc*, *Fos*, and *H1a* show several neural signatures during behavioral tasks. First, catFISH has an anatomical feature of neural activity with cellular levels and identification of cell ensemble activity per unit space of the hippocampus and prefrontal cortex. We found that cell ensemble dynamics in the distal part of CA3 (close to CA2) were recruited more than those in the proximal part of CA3 (close to the dentate gyrus), suggesting that the current conditioning paradigm might be more relevant to contextual effects^{25,34,46,47}. Second, catFISH has a temporal feature for the dynamics of cell ensembles in two distinct events (i.e., Epochs 1 and 2), which were approximately 20 min apart, although neural activity was detected only for two time points. These results showed that Cre⁺ mice had a higher reactivation level of cell ensembles during Epoch 2, which were initially activated during Epoch 1. Third, catFISH indicates a spatiotemporal aspect for dynamic changes between two distinct events in behavioral and cellular levels in multiple brain areas. We showed that freezing behavior had opposite correlations with *Arc*-expressing cell ensembles in the hippocampus and medial prefrontal cortex. Fourth, catFISH compensates these features using different sets of IEG. The present study used two sets of catFISH (i.e., the detection of cytoplasmic *Arc* vs. nuclear *Arc*, and nuclear *H1a* vs. nuclear *Fos*). Then, we obtained two-dimensional results with similar dynamics of cell ensembles in Cre⁻ and Cre⁺ mice. ... These results using catFISH provided evidence for the dynamics of cell ensembles in the hippocampus.”

R1 Q3 (3 of 13).

3. In Figure 4 and Extended Data Fig. 8, there was a higher percentage of Arc+ cells in Cre+ than Cre- mice, indicating that more hippocampal CA3 and CA1 cells were active. Given that Cre+ mice had apnea and reduced memory performance as shown in Fig. 3, it would be helpful if the authors could provide an interpretation and/or discuss the implications of this finding. Is it known whether these Arc+ cells are excitatory or inhibitory? Double staining with specific markers will help to reveal the identity of these neurons.

R1 A3 (1). Thank you very much for the comments of the ideas. We preliminarily tried double staining with *Gad1* and *Arc*, and with *Gad2* and *Arc* in the hippocampus. We could rarely detect *Arc* expression in GABAergic neurons (data not shown) as previously described in Vazdarjanova et al. (2006). According to a recent review article (Giorgi and Marinelli 2021), text has been added for the description about IEG expression on excitatory and inhibitory neurons as follows:

p.13 (Discussion): “Commonly, *Arc* and *H1a* are considered primarily glutamatergic IEGs, whereas *Fos*, *FosB*, *Egr1* (or *zif268*), and *Npas4* are acutely induced by depolarization in GABAergic neurons in culture^{44,45}.”

R1 A3 (2). Moreover, though not directly tested, we assume that a lower percentage of Arc+ cells in Cre- mice could be caused by “inverse synaptic tagging” of Arc protein. Text has been revised as follows:

pp.14-15 (Discussion): “Furthermore, though not directly tested, catFISH might detect a “molecular feature” of states of active and inactive synapses of cell ensembles that previously experienced strong activation. Arc protein is rapidly increased by strong synaptic activity and conversely accumulates in inactive synapses that previously experienced strong activation³⁹. We assume that animals, which previously experienced strong stimuli (i.e., CS+ presentations), could have less accumulation of Arc protein at active synapses of hippocampal cells and more accumulation at inactive synapses (i.e., inverse synaptic tagging³⁹), indicating the decrease of Arc expression during CS- presentations of the recognition session.”

R1 Q4 (4 of 13). *Minor points:*

1. The respiration related signals in the hippocampus could arise from several distinct sources, including the brainstem breathing center and/or sensory feedback via nasal breathing. The current dataset does not prove or rule out either. The authors may want to make their statements more inclusive and consistent. In the first paragraph of introduction, the authors mentioned nasal respiration modulates oscillatory activity in the hippocampus, while in other parts (e.g., abstract), they talked about respiratory signals equivalent to excitatory-to-inspiratory phase transition, implying a brainstem origin.

R1 A3. We appreciate the supportive comments. Text has been added and modified as follows:

p.3 (Introduction): “The breathing oscillators, causing biphasic inspiratory-expiratory rhythm, comprise networking between excitatory and inhibitory neurons in the PreBötzing complex (PreBötC) and Bötzing complex in the medulla oblongata of the brainstem¹⁻⁴. While breathing is altered by activating cognitive processes via the prefrontal cortex in a “top-down” fashion⁵⁻⁷, the respiratory system through the nose play an important role in coupling neural oscillatory activity ...”

R1 Q5 (5 of 13).

2. In the object recognition memory task (Figure 2), was the novel object always placed at the right-side? Should this be randomized to eliminate the potential side preference?

R1 A6. Thank you for the suggestion. According to the reviewer's suggestion, we increased the number of animals (i.e., 13 Cre- mice and 14 Cre+ mice) for the novel object recognition task, and the novel object was placed in a randomized manner.

Methods file, p.4: "The novel object was replaced by one of these objects in a pseudorandom manner."

R1 Q6 (6 of 13).

3. For parametric statistical tests, normal distribution of the datasets should be verified. Otherwise, non-parametric tests should be used.

R1 A6. We greatly appreciate the reviewer state. According to the reviewer's suggestion, all statistical tests have been entirely modified to verify normal distribution using Shapiro-Wilk normality test. We have also modified the graphs (i.e., box plots vs. bar plots) according to the distribution.

Methods pp.13-14: "We tested for normality and sphericity using the Shapiro-Wilk normality test and the Mauchly test for sphericity. We used a two-tailed Welch's *t* test, two-tailed paired *t* test, one-way repeated-measures ANOVA, two-way mixed-design ANOVA, *post hoc* pairwise comparisons using a paired *t* test with Bonferroni correction, a two-tailed one-sample *t* test above zero, and Pearson's product-moment correlation analysis. If the assumptions of normality and sphericity were violated, we performed the nonparametric Mann-Whitney *U* test, Wilcoxon signed-rank test, and Friedman test followed by *post hoc* pairwise comparisons using the Wilcoxon signed-rank test with Bonferroni correction, and applied the Greenhouse-Geisser correction for departure from sphericity.."

R1 Q7 (7 of 13).

4. The phrase "Preliminary experiments with fear conditioning without object strategies" could be simplified to enhance readability.

R1 A7. Text has been modified according to the reviewer's suggestion.

Methods file, p.5: "Preliminary experiments of the fear-conditioning paradigms."

R1 Q8 (8 of 13).

5. Line 104. Please define “RR interval”.

R1 A8. Text has been modified according to the reviewer’s suggestion.

p.5 (Results): “..., **very small changes were found in the R wave-to-R wave (RR) intervals between heartbeats** (1.2-1.3%) measured by electrocardiogram ...”

R1 Q9 (9 of 13).

6. Figure 1. “plethysmography” is misspelled as “pletysmography”.

R1 A9. Text has been modified according to the reviewer’s suggestion.

R1 Q10 (10 of 13).

7. Line 129. “a higher discrimination ratio to zero”. Use “above” instead of “to”.

R1 A10. Since we used two-tailed one-sample *t* test, Text has been modified as follows:

p.6 (Results): “..., Cre⁻ mice had a higher discrimination ratio **against zero (one-sample *t* test)** and ...”

p.23 (Figure Legends): “... , **one sample *t* test against zero**.”

p.23 (Figure Legends): “..., ⁰ *p* < 0.05 (one sample *t* test **against zero**).”

R1 Q11 (11 of 13).

8. Line 243. Anterior cingulate cortex is abbreviated as Cg1. In Figure 4, it is labeled as ACC.

R1 A11. Text has been modified according to the reviewer’s suggestion.

Methods file, p.12: “... was acquired per region of interest in CA3 and CA1 of the hippocampus, and anterior cingulate cortex (**ACC**), ...”

R1 Q12 (12 of 13).

9. Line 259-260. Use “co-expressed” instead of “co-expression”, “were” instead of “was”.

R1 A12. Text has been modified to enhance readability.

Methods file, p.12: “Regarding *Arc* catFISH, we estimated the proportion of cells expressing cytoplasmic and nuclear *Arc*-positive signals and cells coexpressing cytoplasmic and nuclear *Arc*-positive signals (i.e., reactivation of the cells that were initially activated during Epoch 1) ...”

R1 Q13 (13 of 13).

10. Some places could benefit from rephrasing/editing. For example, sentences in Line 67-68 and Line 92-95.

R1 A13. Text has been modified according to the reviewer’s suggestion.

p.3 (Introduction): “Memories are stored through mechanisms of synaptic plasticity that change and endure activity of cell ensembles in the brain, known as memory traces or memory engrams^{16,17}.”

pp.4-5 (Results): “According to Sherman et al.²², optogenetic photostimulation activates inhibitory neurons in the PreBötC, which are specifically expressed with channelrhodopsin-2 (ChR2), and then the inhibitory neurons immediately suppress excitatory neurons in the PreBötC, resulting in respiratory arrest and short-term apnea.”

Reviewer #2 (Remarks to the Author):

... However, I have some critical concerns regarding the interpretation of behavioral experiments. The data does not necessarily point to a memory effect caused by a respiratory signal not arriving at different brain areas. Instead, it could be a general systemic effect of the apnea, a more straightforward explanation. This gets mixed with difficulty following some of the descriptions of the experimental protocols. I explain my concerns more specifically in the points below.

R2 Q1 (1 of 9). Major points

1) I want to focus on the behavioral results of Fig 2b, specifically in the Cre+ group:

First, the discrimination ratio index variability is substantial during the sample epoch. Second, six of nine mice display positive discrimination index values, indicating spending more time with the new object. Third, there are significant changes in the discrimination index between sample and recognition, regardless of the direction of these changes. Fourth, a 2-second period is substantial for a mouse's number of respiratory cycles. A 4-6 Hz respiration rate inhibited for 2 seconds amounts to skipping between 8 and 12 breathing

cycles.

(Statistical tests show differences, but this would happen under several scenarios, especially with this sample size. The fact of the considerable variability is my main concern.)

R2 A1. We greatly appreciate the reviewer state of effect sizes. According to the reviewer's suggestion, we further performed the novel object recognition task and increased the sample size of animals (i.e., Cre⁻ mice: n = 13, Cre⁺ mice: n = 14). All statistical tests have been entirely modified to verify normal distribution using Shapiro-Wilk normality test. If the assumptions of normality and sphericity were violated, we performed the nonparametric tests, and applied the Greenhouse-Geisser correction for departure from sphericity. Regarding Cre⁺ mice, 8 of 14 Cre⁺ mice showed positive discrimination values, while 6 of 14 Cre⁺ mice showed negative values (Fig. 2b-d). Indeed, two variances of the ratio during the sample epoch was not different between Cre⁻ and Cre⁺ mice ($F(13, 12) = 1.33, p > 0.1$). We revised the results of the main text, figure Legends, and Figure 2 as follows:

pp.22-23 (Figure Legends): "**b.** Plots showing the discrimination ratio (to the novel) during the sample and recognition epochs between *Vgat*-Cre⁻ (Cre⁻, n = 13, blue) and *Vgat*-Cre⁺ (Cre⁺, n = 14, orange) mice (animal-type: $F(1, 25) = 3.19, p = 0.09$; epoch: $F(1, 25) = 15.54, p = 0.0006$; interaction: $F(1, 25) = 3.44, p = 0.08$, two-way mixed-design ANOVA). The discrimination ratio was higher during the recognition epoch than during the sample epoch in Cre⁻ mice ($t(12) = 4.45, p = 0.0008$, paired *t* test) but not in Cre⁺ mice ($t(13) = 1.45, p > 0.1$). During the recognition epoch, Cre⁻ mice had a higher discrimination ratio than Cre⁺ mice ($t(19.47) = 2.85, p = 0.01$, Welch's *t* test; Cre⁻: $t(12) = 8.22, p < 0.00001$; Cre⁺: $t(13) = 1.07, p > 0.1$, one-sample *t* test above zero)."

R2 Q2 (2 of 9).

These three points left me wondering about the effect of photostimulation on the mice. If they got their breathing interrupted, the impact of that interruption is likely to be systemic. For example, they could get diminished abilities to perceive, explore, track their position, or understand what is generally happening. Similar to a fleeting moment of dizziness, light-headedness, and/or faintness. And this general state of confusion explains why they vary widely in their behavior. Some of them seem to discriminate, and some others do not, because a transient systemic impact would likely hit differently to different individuals. Therefore, we could have something of this sort instead of a specific effect on memory caused by interrupting the corollary discharge from PreBot to brain structures. So how can we rule out this possibility?

R2 A2. This is a great suggestion. As the reviewer mentioned, we cannot rule out the possibility of a systemic impact on the brain and hit differently to each animal. We entirely revised the

interpretation of the main text as follows:

p.4 (Introduction): "These results demonstrate that the neural signature of hippocampal ensemble dynamics, which may underlie memory encoding and momentary links between information sequences, is stimulated by EI-transition-dependent signals derived from the PreBötC."

p.13 (Discussion): "These findings suggest that components of EI-transition-dependent signals derived from the PreBötC may stimulate a certain regulatory mechanism for the orchestration of CA3 cell ensembles and might be involved in information manipulation rather than memory storage."

p.16 (Discussion): "Thus, we propose that components of inspiratory activity derived from PreBötC (e.g., rates and phases) could be key factors for a certain mechanism of having momentary links between information sequences on a sub-msec temporal scale."

R2 Q3 (3 of 9).

2) I find the description of the design for the fear conditioning experiments confusing. For example, in Figure 3b, the diagram shows that on Day 3, there is no footshock. But, conversely, Extended Data Fig.5 shows a footshock for all groups on Day 3. I might be missing something, but I couldn't understand this by following the descriptions provided in the methods section.

Also, from Figure 3, I understood that all groups received tones as part of the protocol. However, from Extended Data Fig.5, groups A and B did NOT receive tones.

The title of figure 3 reads "Fear conditioning without object effects and optogenetic manipulations", and I don't understand what the authors mean by "object effects". Also, the title seems to indicate there are no optogenetic manipulations.

In general, the rationale for the protocol and the actual sequence of the protocol were quite hard to follow for me. I think the manuscript would benefit from more precise descriptions and more explicit explanations for the design's rationale.

R2 A3. I apologize for the confusion. To maintain the rationale for the protocol of fear conditioning, we further tested additional animals in group G, which performed the task for three days that is the same paradigm for Cre- and Cre+ mice. Text and Figures have been added and modified according to the reviewer's suggestion.

Methods file, p.6: "The animals in Groups A-D performed the fear conditioning task for four days (three for the sample session and one for the recognition session, see Extended Data Fig. 5a,c,e,g), the animals in Groups E and F performed the task for five days (four for the sample

session and one for the recognition session, Extended Data Fig. 7a,b), and the animals in Group G performed the task for three days (two for the sample session and one for the recognition session, Extended Data Fig. 8a).”

p.7 (Results): “Several conditioning paradigms were preliminarily tested in combination with a box, contexts, lights, and tones (i.e., Groups A-G, see Methods, Extended Data Figs. 4-8). We found that animals in the acrylic box effectively conditioned a combination of context and tones, which were associated with footshocks, and used the paradigm of Group G for further experiments (modified from the paradigm of Group C, i.e., CS⁻: black context and low tones; CS⁺: white context and high tones, see Extended Data Fig. 8).

Then, we examined whether short-term PreBötC-induced apnea occurring during encoding decreased subsequent memory performance.”

R2 Q4 (4 of 9).

But even after understanding the protocol, I see the same issue I described for the object recognition task. Again, there is considerable variability across individuals for the % active time (Fig 3 h, i). Half of Cre⁻ mice seem to decrease their active time during no-tone periods. In Cre⁺, half of the group reduces their active time, and the other half increments it. Variability is again quite significant for the tone period in the Cre⁺ mice. The wide behavioral variation suggests that the mice might be confused instead of having a specific memory problem. It is hard to decide what is going on with this sample size.

R2 A4. Thank you very much for the state of results of the conditioning task. According to the reviewer’s suggestion, we further performed the novel object recognition task to increase the sample size of animals (i.e., Cre⁺ mice: n = 8). Then, in Cre⁻ and Cre⁺ mice during the recognition session, the Shapiro-Wilk normality test showed that the active time in each CS⁻ and CS⁺ was normally distributed. We did not have different variances of active time between Cre⁻ and Cre⁺ mice (offset during Epoch 1: $F(7, 5) = 0.48$, $p > 0.1$; offset during Epoch 2: $F(7, 5) = 0.95$, $p > 0.1$; onset during Epoch 1: $F(7, 5) = 1.20$, $p > 0.1$, onset during Epoch 2: $F(7, 5) = 1.08$, $p > 0.1$).

R2 Q5 (5 of 9).

3) The dynamics and correlations found in the expression of Arc can be interpreted in several ways, given the possibility of systemic effects described earlier. The evidence provided by the expression of this IEG does not solve or address the issues raised earlier (points 1 and 2).

R2 A5. We greatly appreciate the suggestion for the interpretation of the results and the current

technique. We totally agree that we cannot rule out the possibility of systematic effects on the brain. However, the catFISH approach still has several benefits to determine dynamic changes of cell ensembles in the hippocampal CA1 and CA3 as well as the prefrontal cortex (e.g., Nomoto et al., 2016, *nat commun* 7:12319; Sakaguchi et al., 2018, *nat commun* 9:3526; Suzuki et al., 2022, *nat commun.* 13:41). The current results in IEG catFISH show the possibility that “the neural signature of hippocampal ensemble dynamics, which may underlie memory encoding and momentary links between information sequences, is stimulated by EI-transition-dependent signals derived from the PreBötC.” (p.4).

We have further performed the other set of IEG catFISH for nuclear *Fos* and nuclear *Homer 1a*, which IEGs are also confirmed to determine the dynamics of cell ensembles (Bottai et al., 2001; Vazdarjanova and Guzowski 2004; Chawla et al., 2004; Nakamura and Sauvage 2016). We revised the description of this approach as follows:

pp.13-14 (Discussion): “[The catFISH techniques for *Arc*, *Fos*, and *H1a* show several neural signatures during behavioral tasks. First, catFISH has an anatomical feature of neural activity with cellular levels and identification of cell ensemble activity per unit space of the hippocampus and prefrontal cortex. We found that cell ensemble dynamics in the distal part of CA3 \(close to CA2\) were recruited more than those in the proximal part of CA3 \(close to the dentate gyrus\), suggesting that the current conditioning paradigm might be more relevant to contextual effects^{25,34,46,47}. Second, catFISH has a temporal feature for the dynamics of cell ensembles in two distinct events \(i.e., Epochs 1 and 2\), which were approximately 20 min apart, although neural activity was detected only for two time points. These results showed that Cre⁺ mice had a higher reactivation level of cell ensembles during Epoch 2, which were initially activated during Epoch 1. Third, catFISH indicates a spatiotemporal aspect for dynamic changes between two distinct events in behavioral and cellular levels in multiple brain areas. We showed that freezing behavior had opposite correlations with *Arc*-expressing cell ensembles in the hippocampus and medial prefrontal cortex. Fourth, catFISH compensates these features using different sets of IEG. The present study used two sets of catFISH \(i.e., the detection of cytoplasmic *Arc* vs. nuclear *Arc*, and nuclear *H1a* vs. nuclear *Fos*\). Then, we obtained two-dimensional results with similar dynamics of cell ensembles in Cre⁻ and Cre⁺ mice. Furthermore, though not directly tested, catFISH might detect a “molecular feature” of states of active and inactive synapses of cell ensembles that previously experienced strong activation. *Arc* protein is rapidly increased by strong synaptic activity and conversely accumulates in inactive synapses that previously experienced strong activation³⁹. We assume that animals, which previously experienced strong stimuli \(i.e., CS⁺ presentations\), could have less accumulation of *Arc* protein at active synapses of hippocampal cells and more accumulation at inactive synapses \(i.e., inverse synaptic tagging³⁹\), indicating the](#)

decrease of *Arc* expression during CS⁻ presentations of the recognition session. These results using catFISH provided evidence for the dynamics of cell ensembles in the hippocampus.”

R2 Q6 (6 of 9). *Minor points.*

Figure 1e,1f. Please provide individual data points

R2 A6. Thank you very much for the comments. Since the assumptions of normality was violated, we reanalyzed the respiratory frequency and RR intervals under isoflurane-induced anesthesia. Figure 1 and text have been modified according to the reviewer’s suggestion.

pp.21-22 (Figure legends): “**d-f.** Plots showing the respiratory frequency (**d,e**) and RR intervals of electrocardiogram (**d,f**) during the prestimulation period (2 s), photostimulation (flat, 465 nm as blue light, 2 s), and the poststimulation period (2 s) in anesthetic *Vgat-Cre*⁺ mice (20 trials per animal, n = 4; respiratory frequency: $\chi^2(2) = 120.33$, $p < 0.00001$, Friedman test; photostimulation: $p < 0.00001$ compared to prestimulation, $p < 0.00001$ compared to poststimulation, *post hoc* pairwise Wilcoxon signed-rank test with Bonferroni correction; RR interval: $\chi^2(2) = 68.03$, $p < 0.00001$; prestimulation: $p < 0.00001$ compared to photostimulation, $p < 0.00001$ compared to poststimulation).”

R2 Q7 (7 of 9). *Methods:*

line 18. "was" should be "as"

R2 A7. Text has been modified according to the reviewer’s suggestion

Methods file, p.1: “... were produced by using the AAV Helper Free Expression System (Cell Biolabs, Inc., San Diego, CA) as previously described⁴⁴.”

R2 Q8 (8 of 9).

line 142. it says, "In the recognition session (day 4)". I thought recognition sessions occurred on day 3.

R2 A8. I apologize for the confusion. We further tested additional animals in Group G, which performed the task for 3 days. Text has been added to explain about the days of the task in each group of animals.

Methods file, p.6: “The animals in Groups A-D performed the fear conditioning task for four days”

(three for the sample session and one for the recognition session, see Extended Data Fig. 5a,c,e,g), the animals in Groups E and F performed the task for five days (four for the sample session and one for the recognition session, Extended Data Fig. 7a,b), and the animals in Group G performed the task for three days (two for the sample session and one for the recognition session, Extended Data Fig. 8a).”

R2 Q9 (9 of 9).

Fig3. There is no mention of the groups (A, B, C, D) in the figure, which are essential for understanding the design. This information should be added to the figure.

To which group do the behavioral results belong?

R2 A9. Text has been added and modified according to the reviewer’s suggestion.

p.7 (Results): “Several conditioning paradigms were preliminarily tested in combination with a box, contexts, lights, and tones (i.e., Groups A-G, see Methods, Extended Data Figs. 4-8). We found that animals in the acrylic box effectively conditioned a combination of context and tones, which were associated with footshocks and used the paradigm of Group G for further experiments (modified from the paradigm of Group C, i.e., CS⁻: black context and low tones; CS⁺: white context and high tones, see Extended Data Fig. 8).

Then, we examined whether short-term PreBötC-induced apnea occurring during encoding decreased subsequent memory performance.”

REVIEWER COMMENTS

Reviewer #1 (Remarks to the Author):

The authors conducted new experiments and analysis for the revision. In addition to the original apnea experiments, now they showed the effects of introducing irregularity in respiration during the time of encoding. The authors also expanded the catFISH (detection of cytoplasmic versus nuclear immediate early gene expression) experiments in subregions of the hippocampus. Manipulating respiration rate and phase in behaving mice is technically challenging and this study does represent one of first attempts on this front. However, similar concerns on interpretation of the behavioral data still remain.

The main claim of the paper is that the rate and phase of the central inspiratory activity contributes to hippocampal memory encoding. In the new experiments (Fig. 6), 10 Hz photostimulation did not change the breathing rate but altered the phase. Behaviorally, this manipulation led to better performance of the Cre+ mice than Cre- control mice (Fig. 6g). The authors seemed to interpret this as a recovery from apnea-induced impairment of memory encoding. There are several issues about this new experiment.

1. It is not clear why 10 Hz photostimulation (phase irregularity) led to better performance of Cre+ mice than control mice. This finding suggests that this manipulation does something more than “rescuing” apnea-induced impairment. Of all the manipulations, the breathing rate is either close to normal or zero (apnea, which may have other systematic effects), the conclusion that the breathing rate contributes to memory encoding seems an overstatement.
2. The rationale of using 10 Hz photostimulation is not very clear. Can the authors manipulate the breathing rate to an intermediate value? For instance, mice breathe at around 4 Hz when freezing during fear retrieval. Testing an intermediate breathing rate would be informative.
3. In Fig. 6a, 10 Hz photostimulation also changed the respiration amplitude towards the second half. Is this a consistent finding?
4. It would be helpful to demonstrate how the phase change, “an increase of 12.3-14.2% (44.4-51.6° per 360° of single respiratory cycle)”, is determined and quantified.

Minor points:

1. It would be helpful to report how flat photostimulation (apnea) and 10 Hz photostimulation change the mouse behavior.
2. Line 280. Fig. 6e, f should be Fig. 6f, g. Fig. 6 figure legend (a-f) does not match its panels (a-g).

3. Line 296. "12.3-14.3% CV". Should it be "12.3-14.3% increase in CV of respiratory wavelength"? Could the authors provide a definition for "respiratory wavelength"?

4. Line 365-367. "Since averaged respiratory frequency in Cre+ mice was not altered under this manipulation (5.9-7.2 Hz), it is likely that a respiratory rate might be a clue component to modulate the coordination with cognitive function." This statement is confusing. If the respiratory frequency is not changed, how could the respiratory rate contribute as stated?

5. Line 550. "anesthasic" should be "anesthetic".

6. Extended Data Fig. 8 figure legend. Panel c: The animals in group G (n=4) learned white walls/high tones as the CS- and black walls/low tones as the CS+ (see Methods). This seems the opposite than what is shown in the figure, where white walls/high tones are paired with foot shock, should be CS+.

7. Extended Data Fig. 8 panel d: "a scale bar is 2s in d". There is no scale bar.

Reviewer #2 (Remarks to the Author):

The authors have adequately addressed most of my previous comments and questions. I only have one remaining comment regarding my previous question #2.

R2Q2.

The statements in the response do not refer to or mention the likely systemic effect of the experimental intervention (photostimulation), which is a key component in interpreting the results.

In the manuscript, they include a line about this (lines 297-298), but without a discussion. What do they have to offer as an argument against a systemic effect explaining these results?

Especially considering:

-that there are cardiovascular effects of the photostimulation.

-the important variability in behavioral results (ie fig2b), which are the foundation of the claim that the intervention alters memory encoding.

Response to Reviews' comments

NCOMMS-22-07329A-Z

Hippocampal ensemble dynamics modulated by respiration during memory encoding

Corresponding Author: Nozomu H. Nakamura

According to reviewers' suggestion, the number of animals was increased to determine more precise optogenetic effects on behavioral performance using the manipulation with the following photostimulation: i) flat photostimulation (Cre- mice: n = 11, Cre+ mice: n = 11); ii) 10-Hz photostimulation (Cre- mice: n = 10, Cre+ mice: n = 8); and iii) 4-Hz photostimulation (Cre- mice: n = 9, Cre+ mice: n = 11). We have again marked the corrected words and sentences with "blue" in the manuscript.

Reviewer #1 (Remarks to the Author):

... Manipulating respiration rate and phase in behaving mice is technically challenging and this study does represent one of first attempts on this front. However, similar concerns on interpretation of the behavioral data still remain.

R1 Q1 (1 of 11).

The main claim of the paper is that the rate and phase of the central inspiratory activity contributes to hippocampal memory encoding. In the new experiments (Fig. 6), 10 Hz photostimulation did not change the breathing rate but altered the phase. Behaviorally, this manipulation led to better performance of the Cre+ mice than Cre- control mice (Fig. 6g). The authors seemed to interpret this as a recovery from apnea-induced impairment of memory encoding. There are several issues about this new experiment.

1. It is not clear why 10 Hz photostimulation (phase irregularity) led to better performance of Cre+ mice than control mice. This finding suggests that this manipulation does something more than "rescuing" apnea-induced impairment. Of all the manipulations, the breathing rate is either close to normal or zero (apnea, which may have other systematic effects), the conclusion that the breathing rate contributes to memory encoding seems an overstatement.

R1 A1. We appreciate the reviewer's suggestion regarding the interpretation of results of manipulating respiratory rates at 10-Hz photostimulation in behaving mice. According to this suggestion, text has been modified to explain as follows:

--

p.13 (Results): “Our results indicated that the irregular rhythmic patterns of PreBötC-induced activity during encoding enhanced memory performance during the whole CS⁺ presentation (i.e., with and without tones).”

p.16 (Discussion): “Optogenetic manipulation with 10-Hz photostimulation induced irregular breathing cycles at the exact time of encoding, and then led to better memory performance. This finding could be supported by the theory of stochastic resonance³⁵ that the irregular signal inputs may improve hippocampal-dependent memory³⁶.”

R1 Q2 (2 of 11).

2. The rationale of using 10 Hz photostimulation is not very clear. Can the authors manipulate the breathing rate to an intermediate value? For instance, mice breathe at around 4 Hz when freezing during fear retrieval. Testing an intermediate breathing rate would be informative.

R1 A2. We greatly appreciate the positive suggestion for an additional experiment. According to the reviewer’s suggestion, we performed the further experiment of the conditioning paradigm using 4-Hz photostimulation (Cre⁻ mice: n = 9; Cre⁺ mice: n = 11). Cre⁺ mice breathed at 3.35 ± 0.21 Hz under the photostimulation. Text has been added to explain the results and discussion regarding the additional experiment.

--

pp.13-15 (Results): “**Fear conditioning using optogenetic manipulation and 4-Hz photostimulation.** In the conditioning experiment with photostimulation at 4 Hz, the frequency of whole-body plethysmographic signals in Cre⁺ mice was decreased as 0.51 Hz (median) during flat photostimulation (see Fig. 7a,b), and 3.35 Hz (mean) during photostimulation at 4 Hz (162.5-ms pulse-on and 87.5-ms pulse-off, duty cycle: 65.0%; Cre⁺: $p = 0.038$, Mauchly tests for sphericity; $F(2, 20) = 331.1$, $p < 0.00001$ with Greenhouse-Geisser correction; 4-Hz photostim.: $p < 0.00001$ compared to prestimulation, $p < 0.00001$ compared to poststimulation, *post hoc* pairwise t test with Bonferroni correction; Cre⁻: $F(2, 16) = 0.42$, $p > 0.1$; Cre⁻ vs. Cre⁺ in 4-Hz photostim.: $t(10.53) = 15.96$, $p < 0.00001$, Welch’s t test, Fig. 7c). Furthermore, 4-Hz photostimulation did not change the variability in the cycle duration in Cre⁻ and Cre⁺ mice (animal type: $F(1, 18) = 2.14$, $p > 0.1$, period: $F(1, 18) = 1.81$, $p > 0.1$; interaction: $F(1, 18) = 0.03$, $p > 0.1$, two-way mixed-design ANOVA, Fig. 7d). Although changes in the pressure amplitude of whole-body plethysmographic signals between the prestimulation and 4-Hz photostimulation periods were below zero in both Cre⁻ and Cre⁺ mice, Cre⁺ mice had lower percentage changes than Cre⁻ mice (Cre⁻ vs. Cre⁺: $t(14.61) = 5.21$, $p = 0.0001$, Welch’s t test; Cre⁻: $t(8) = 2.51$, $p = 0.036$; Cre⁺: $t(10) = 7.12$, $p = 0.00003$, one-sample t test against zero, Fig. 7e).

In the conditioning task, no difference was observed in the activity levels of Cre⁻ and Cre⁺ mice during the footshock in the sample sessions (see Extended Data Fig. 12c,d). Regarding the discrimination ability of Cre⁺ mice, no difference in the active time was observed between the CS⁻ and CS⁺ presentations during the onset time blocks (offset: animal type: $F(1, 18) = 1.09, p > 0.1$; epoch: $F(1, 18) = 5.99, p = 0.02$; interaction: $F(1, 18) = 0.33, p > 0.1$; CS⁻ vs. CS⁺: Cre⁻: $t(8) = 1.79, p > 0.1$; Cre⁺: $t(10) = 1.68, p > 0.1$; onset: animal type: $F(1, 18) = 1.56, p > 0.1$; epoch: $F(1, 18) = 7.81, p = 0.012$; interaction: $F(1, 18) = 4.20, p = 0.055$; CS⁻ vs. CS⁺: Cre⁻: $t(8) = 3.16, p = 0.013$; Cre⁺: $t(10) = 0.75, p > 0.1$, Fig. 7f,g). Strikingly, the active time during CS⁻ presentation was lower in Cre⁻ mice than in Cre⁺ mice (Cre⁻ vs. Cre⁺ in CS⁻: $t(18.00) = 2.42, p = 0.03$, Fig. 7g). These results revealed that reducing the frequency of PreBötC-induced activity at the exact time of encoding caused a decline in memory performance.”

p.16 (Discussion): “Moreover, respiratory activity induced by photostimulation at a lower frequency (i.e., 4 Hz) during encoding caused ambiguous behavior and reduced the ability of Cre⁺ mice to discriminate between CS⁻ and CS⁺ presentations, considering a different type of memory impairments (i.e., a type I error or false positive occurring during CS⁻ presentation). Cre⁺ mice slowed down during flat photostimulation and photostimulation at different frequencies, but their behavior looked natural and healthy afterwards. It is likely that the frequency and patterns of central respiratory activity might be key drivers in modulating momentary links between information sequences on the order of 100 milliseconds. The manipulation of PreBötC-induced activity may have specific effects on associative memory that could be caused by either interrupting or enhancing the corollary discharge from the PreBötC to memory structures in the hippocampus¹⁸.”

R1 Q3 (3 of 11).

3. In Fig. 6a, 10 Hz photostimulation also changed the respiration amplitude towards the second half. Is this a consistent finding?

R1 A3. Thank you for the comments for data of the respiratory amplitude. According this suggestion, we analyzed and showed changes in the pressure amplitude of respiratory cycles using by 10 Hz and 4 Hz photostimulation (Figs. 6e and 7e).

--

pp.12-13 (Results): “Additionally, in Cre⁺ mice, the pressure amplitude of whole-body plethysmographic signals was decreased from the prestimulation to 10-Hz photostimulation periods (Cre⁻ vs. Cre⁺: $t(13.20) = 6.91, p < 0.00001$, Welch’s t test; Cre⁻: $t(9) = 0.13, p > 0.1$; Cre⁺: $t(7) = 8.37, p = 0.00007$, one-sample t test against zero, Fig. 6e).”

p.14 (Results): “Although changes in the pressure amplitude of whole-body plethysmographic signals between the prestimulation and 4-Hz photostimulation periods were below zero in both Cre⁻ and Cre⁺ mice, Cre⁺ mice had lower percentage changes than Cre⁻ mice (Cre⁻ vs. Cre⁺: $t(14.61) = 5.21$, $p = 0.0001$, Welch’s t test; Cre⁻: $t(8) = 2.51$, $p = 0.036$; Cre⁺: $t(10) = 7.12$, $p = 0.00003$, one-sample t test against zero, Fig. 7e).”

R1 Q4 (4 of 11).

4. *It would be helpful to demonstrate how the phase change, “an increase of 12.3-14.2% (44.4-51.6° per 360° of single respiratory cycle)”, is determined and quantified.*

R1 A4. We apologize for the confusion how the phase change is. To exclude the confusion, we removed the description of “44.4-51.6° per 360° of single respiratory cycle”, and added to explain the definition of coefficient of variation (CV). Of note, the number of animals was increased in the experiment using 10 Hz photostimulation (Cre⁻ mice: $n = 10$, Cre⁺ mice: $n = 8$).

--

pp.3-4 in Methods file: “The variability in the animal’s respiratory activity in the acrylic box was calculated by the coefficient of variation (CV) of the cycle duration of the whole-body plethysmographic signals. The CV was defined as the ratio of the standard deviation to the mean. Although whole-body plethysmographic signals included small physical movement signals in awake rodents, their frequency and cycle duration almost certainly reflected the frequency and cycle duration of respiration⁶².”

p.12 (Results): “Moreover, the variability in the respiratory activity was quantified by the coefficient of variation (CV) of the cycle duration of whole-body plethysmographic signals (see Methods). The results showed that 10-Hz photostimulation increased CV of the cycle duration by 14.6-15.4% in Cre⁺ mice compared to that in Cre⁻ mice (animal type: $F(1, 16) = 3.13$, $p = 0.096$, period: $F(1, 16) = 15.10$, $p = 0.0013$; interaction: $F(1, 16) = 22.81$, $p = 0.0002$, two-way mixed-design ANOVA; prestimulation vs. 10-Hz photostim.: Cre⁻: $t(9) = 0.60$, $p > 0.1$; Cre⁺: $t(7) = 4.36$, $p = 0.003$, paired t test; Cre⁻ vs. Cre⁺ in 10-Hz photostim.: $t(13.72) = 3.03$, $p = 0.0092$, Welch’s t test, Fig. 6d).”

R1 Q5 (5 of 11). *Minor points:*

1. *It would be helpful to report how flat photostimulation (apnea) and 10 Hz photostimulation change the mouse behavior.*

R1 A5. Thank you for the suggestion. Text has been added to mention mouse behavior manipulating photostimulation

--

p.16 (Discussion): “Cre⁺ mice slowed down during flat photostimulation and photostimulation at different frequencies, but their behavior looked natural and healthy afterwards.”

R1 Q6 (6 of 11).

2. Line 280. Fig. 6e, f should be Fig. 6f, g. Fig. 6 figure legend (a-f) does not match its panels (a-g).

R1 A6. We apologize for the confusion. Text has been modified according to the reviewer’s suggestion.

R1 Q7 (7 of 11).

3. Line 296. “12.3-14.3% CV”. Should it be “12.3-14.3% increase in CV of respiratory wavelength”? Could the authors provide a definition for “respiratory wavelength”?

R1 A7. We apologize for the confusion. To avoid the confusion, we used the description of “the cycle duration of respiration” instead of “respiratory wavelength”. And text has been added to explain about the variability of respiration and the definition of CV:

--

pp.3-4 in Methods file: “The variability in the animal’s respiratory activity in the acrylic box was calculated by the coefficient of variation (CV) of the cycle duration of the whole-body plethysmographic signals. The CV was defined as the ratio of the standard deviation to the mean. Although whole-body plethysmographic signals included small physical movement signals in awake rodents, their frequency and cycle duration almost certainly reflected the frequency and cycle duration of respiration⁶².”

p.12 (Results): “Moreover, the variability in the respiratory activity was quantified by the coefficient of variation (CV) of the cycle duration of whole-body plethysmographic signals (see Methods). The results showed that 10-Hz photostimulation increased CV of the cycle duration by 14.6-15.4% in Cre⁺ mice compared to that in Cre⁻ mice (animal type: $F(1, 16) = 3.13, p = 0.096$, period: $F(1, 16) = 15.10, p = 0.0013$; interaction: $F(1, 16) = 22.81, p = 0.0002$, two-way mixed-design ANOVA; prestimulation vs. 10-Hz photostim.: Cre⁻: $t(9) = 0.60, p > 0.1$; Cre⁺: $t(7) = 4.36, p = 0.003$, paired t test; Cre⁻ vs. Cre⁺ in 10-Hz photostim.: $t(13.72) = 3.03, p = 0.0092$, Welch’s t test, Fig. 6d).”

R1 Q8 (8 of 11).

4. Line 365-367. “Since averaged respiratory frequency in Cre⁺ mice was not altered under this manipulation (5.9-7.2 Hz), it is likely that a respiratory rate might be a clue component to modulate the coordination with

cognitive function.” This statement is confusing. If the respiratory frequency is not changed, how could the respiratory rate contribute as stated?

R1 A8. We apologize the confusion. This explanation has been removed. Then, text has been added to explain about the effect of different frequency of photostimulation during encoding.

--

pp.15-16 (Discussion): “This study showed that online encoding is modulated by central respiratory activity. Importantly, memory performance is improved or deteriorated by the frequency and regularity of PreBötC-induced activity during encoding. In the conditioning task, optogenetic manipulation with flat photostimulation (i.e., apnea) during encoding did not change behavior, in which Cre⁺ mice exhibited a type II error or false negative during CS⁺ presentation. Optogenetic manipulation with 10-Hz photostimulation induced irregular breathing cycles at the exact time of encoding, and then led to better memory performance. This finding could be supported by the theory of stochastic resonance³⁵ that the irregular signal inputs may improve hippocampal-dependent memory³⁶. Moreover, respiratory activity induced by photostimulation at a lower frequency (i.e., 4 Hz) during encoding caused ambiguous behavior and reduced the ability of Cre⁺ mice to discriminate between CS⁻ and CS⁺ presentations, considering a different type of memory impairments (i.e., a type I error or false positive occurring during CS⁻ presentation). Cre⁺ mice slowed down during flat photostimulation and photostimulation at different frequencies, but their behavior looked natural and healthy afterwards. It is likely that the frequency and patterns of central respiratory activity might be key drivers in modulating momentary links between information sequences on the order of 100 milliseconds. The manipulation of PreBötC-induced activity may have specific effects on associative memory that could be caused by either interrupting or enhancing the corollary discharge from the PreBötC to memory structures in the hippocampus¹⁸.”

R1 Q9 (9 of 11).

5. Line 550. “anesthasic” should be “anesthetic”.

R1 A9. Thank you for noticing it. Text has been modified.

R1 Q10 (10 of 11).

6. Extended Data Fig. 8 figure legend. Panel c: The animals in group G (n=4) learned white walls/high tones as the CS- and black walls/low tones as the CS+ (see Methods). This seems the opposite than what is shown in the figure, where white walls/high tones are paired with foot shock, should be CS+.

R1 A10. We appreciate the reviewer’s comment. Text has been modified as follows:

--

Figure legends in Extended Data Fig.8: "The animals in group G (n = 4 males) learned white walls/high tones as the CS⁺ and black walls/low tones as the CS⁻ (see Methods)."

R1 Q11 (11 of 11).

7. Extended Data Fig. 8 panel d: "a scale bar is 2s in d". There is no scale bar.

R1 A11. Thank you for noticing it. Text has been removed.

Reviewer #2 (Remarks to the Author):

The authors have adequately addressed most of my previous comments and questions. I only have one remaining comment regarding my previous question #2.

R2Q1 (1 of 1).

The statements in the response do not refer to or mention the likely systemic effect of the experimental intervention (photostimulation), which is a key component in interpreting the results. In the manuscript, they include a line about this (lines 297-298), but without a discussion. What do they have to offer as an argument against a systemic effect explaining these results?

Especially considering:

-that there are cardiovascular effects of the photostimulation.

-the important variability in behavioral results (ie fig2b), which are the foundation of the claim that the intervention alters memory encoding.

R2 R1. We greatly appreciate the important suggestion. According to the reviewer's suggestion, we have stated the explanation of a systemic effect in discussion.

--

pp.16-17 (Discussion): "However, we cannot rule out the possibility that manipulation of PreBötC-induced apnea and degraded frequencies causes transient systemic impacts throughout the whole brain. Indeed, cardiovascular effects were observed in Cre⁺ mice, and these mice might get diminished abilities to perceive, explore, and understand what was happening, which could be similar to fleeting moments of dizziness, light-headedness, or faintness. Thus, transient systemic impacts may affect individual Cre⁺ mice differently. More refined *in vivo* electrophysiological recordings from the cell ensembles and brain regions would be essential to

elucidate the systemic effects or network pathways derived from the PreBötC.”

REVIEWERS' COMMENTS

Reviewer #2 (Remarks to the Author):

The authors have adequately addressed my comment. I have no further questions or unresolved issues to raise.